



# Assessment of the spectral downward irradiance at the surface of the Mediterranean Sea using the OASIM ocean-atmosphere radiative model

Paolo Lazzari[1], Stefano Salon[1], Elena Terzić[1], Watson W. Gregg[2], Fabrizio D'Ortenzio[3], Vincenzo Vellucci[4], Emanuele Organelli[3,5] and David Antoine[6,3]

[1]Oceanography Section, OGS – National Institute for Oceanography and Applied Geophysics, Via Beirut 2-4, 34151, Trieste, Italy

[2]Global Modeling and Assimilation Office, NASA/Goddard Space Flight Center, Greenbelt, MD 20771, United States

[3]Sorbonne Université, CNRS, Laboratoire d'Océanographie de Villefranche, LOV, F-06230, Villefranche-sur-Mer, France

[4]Sorbonne Université, CNRS, Institut de la Mer de Villefranche, IMEV, F-06230, Villefranche-sur-Mer, France

[5]Consiglio Nazionale delle Ricerche (CNR), Istituto di Scienze Marine (ISMAR), Via Fosso del Cavaliere 100, 00133 Rome, Italy

[6]Remote Sensing and Satellite Research Group, School of Earth and Planetary Sciences, Curtin University, Perth, WA 6845, Australia

*Correspondence to*: Paolo Lazzari (plazzari@inogs.it)

**Abstract.** A multiplatform assessment of the Ocean–Atmosphere Spectral Irradiance Model (OASIM) radiative model focussed on the Mediterranean Sea is presented. BOUée pour l'acquiSition d'une Série Optique à Long termE (BOUSSOLE) mooring and biogeochemical Argo (BGC-Argo) float optical sensor observations are combined with model outputs to analyse

the spatial and temporal variabilities in the downward planar irradiance at the ocean-atmosphere interface. The correlations (r) between the data and model are always higher than 0.6. At the scale of the BOUSSOLE sampling (15-minute temporal resolution), the root mean square difference (RMSD) oscillates at approximately 30~40% of the averaged model output and is reduced to approximately 10% when the daily variability is filtered out. Both BOUSSOLE and BGC-Argo indicate that the bias is high for the irradiance at 412 nm, whereas it decreases to less than 5% at the other wavelengths. Analysis of atmospheric

input data indicates that the model skill is strongly affected by cloud dynamics and seasonality. High skills are observed during summer when the cloud cover is low.

# 1 Introduction

The availability of in situ oceanic radiometric data has recently increased owing to the deployment of autonomous robotic profiling platforms called biogeochemical Argo floats (hereafter referred to as BGC-Argo floats; Johnson and Claustre, 2016)



equipped with radiometric sensors (Organelli et al., 2016). Such data may be notably exploited to improve the calibration and tuning of the bio-optical models embedded in three-dimensional global and regional physical-BGC coupled models, which will further increase with the development of new autonomous profiling floats dedicated to ocean colour measurements (Leymarie et al., 2018) and the growing observational opportunities represented by the Ocean Colour Radiometry Virtual Constellation (OCR-VC) satellite. In turn, these new observations will pave the way towards the next generation of ocean

biogeochemical models. State-of-the-art bio-optical algorithms, with the atmospheric component providing multispectral light boundary conditions at the sea-water interface with different levels of complexity, are already integrated into the biogeochemical models developed at the Massachusetts Institute of Technology (MIT, Dutkiewicz et al., 2015), Commonwealth Scientific and Industrial Research Organisation (CSIRO, Baird et al., 2016) and National Aeronautics and Space Administration (NASA, Gregg and Rousseaux, 2017). Additionally, the direct assimilation of optical/radiometric data

(Jones et al., 2016) yields a higher robustness than does traditional assimilation of modelled optical/radiometric data based on the chlorophyll-a concentration due to the greater in-depth knowledge of the uncertainties in optical measurements (Dowd et al., 2014).

In this framework, the Mediterranean Sea appears to be a key area for the development of a new multispectral bio-optical model to be potentially integrated with the European Copernicus Marine Environment Monitoring Service (CMEMS). A dense

network of BGC-Argo floats providing quality-controlled radiometric data has indeed been deployed in the Mediterranean Sea (Organelli et al., 2016, Organelli et al. 2017), which can be combined with the high-frequency radiometric and bio-optical observations acquired by a dedicated fixed platform (BOUSSOLE; Antoine et al., 2006, 2008). This meets the requirements and high data quality standard expected for remote system calibration of ocean colour spaceborne sensors (Antoine et al., 2020) and the Copernicus biogeochemical operational model system (MedBFM; Lazzari et al., 2010, 2012, 2016; Cossarini et

al., 2015; Teruzzi et al., 2014, 2018, 2019; Salon et al., 2019), thus realizing analysis, forecasting and reanalysis of the biogeochemical state by the CMEMS, recently upgraded to assimilate BGC-Argo float data (Cossarini et al., 2019).

Assessment of the quality of the direct and diffuse downward irradiance produced by the multispectral Ocean-Atmosphere Spectral Irradiance Model (OASIM; Gregg and Casey, 2009) at the sea surface is a prerequisite to constrain bio-optical in-water light propagation modelling. This activity has been carried out within the framework of the CMEMS Service Evolution

project BIOPTIMOD, which is aimed at the development of a new multispectral bio-optical model that will include the integration of MedBFM with data provided by both BGC-Argo floats and multispectral satellite sensors (e.g., the Ocean and Land Colour Instrument, OLCI, on-board Sentinel3-A and Sentinel3-B; Donlon et al, 2012).

In MedBFM, the computation of the downward irradiance at the sea surface will be updated with OASIM, which has been pre-operationally interfaced with ECMWF atmospheric products and validated against reference data in the Mediterranean

Sea, provided by the BOUSSOLE buoy and BGC-Argo floats.

This work also contributes to the extension of the validation of OASIM in the Mediterranean Sea, an area not covered by the previous skill assessment of Gregg and Casey (2009).



Section 2 presents the OASIM model, the required input data and the datasets adopted for validation purposes. The results are provided in Section 3 and examined in Section 4. Conclusions are summarized in Section 5.


## 2 Materials and Methods

### 2.1 OASIM model

OASIM (Gregg and Casey, 2009; hereafter referred to as GC2009) simulates the propagation of the downward spectral
radiance in the atmosphere and provides the direct and diffuse irradiance over the ocean surface as the output (Fig. 1) at 33 wavelengths ranging from 200 nm to 4 μm (15 wavelengths at a 25-nm spectral resolution in the near-ultraviolet (UV) and visible regions of the light spectrum, 350 - 700 nm).

OASIM is currently applied in several ocean general circulation models, such as in Poseidon (Gregg, 2000; Gregg and Casey, 2007; Rousseaux and Gregg, 2015) and MOM4 (Gregg and Rousseaux, 2016) of the NASA Global Modelling and
Assimilation Office (GMAO), HYCOM (Romanou et al., 2013, 2014) and Russell (Romanou et al., 2014) of the NASA Goddard Institute for Space Studies (GISS) and the MIT OGCM (Dutkiewicz et al., 2015).

OASIM is based on the RADTRAN spectral model developed by Gregg and Carder (1990) for clear-sky conditions, with an upgraded aerosol parameterization scheme, and on the Slingo (1989) parameterization scheme for the spectral cloud transmittance and considers the spectral absorption and scattering of atmospheric gases (ozone, water vapour, oxygen, and
carbon dioxide). All model parameters (extra-terrestrial solar irradiance, Rayleigh optical thickness and absorption coefficients for the various atmospheric gases) are detailed for each of the 33 bands in Tab. 2 of GC2009.

In OASIM, gaseous absorption by ozone, oxygen, carbon dioxide and water vapor is resolved before cloud transmittance determination, while aerosol effects are ignored in the presence of clouds. In the clear-sky parameterization scheme, the role of aerosols is described by three parameters: the aerosol optical thickness (AOT), single scattering albedo and asymmetry.
These quantities have been applied in GC2009 exploiting Moderate Resolution Imaging Spectroradiometer (MODIS; Remer et al., 2005) data at 7 wavelengths ranging from 470 nm to 2.13 μm (for the out-of-range wavelengths, linear extrapolation is performed) from February 2000 to July 2007, extended to 2017 in the present work.

The Slingo model requires four cloud properties as input: cloud cover, $cov$ [%], cloud liquid water path, $LWP$ [g m$^{-2}$] (ice clouds are not considered in OASIM), spectral cloud optical thickness, $\tau_c(\lambda)$ [dimensionless], and cloud droplet effective
radius, $r_e$ [μm]. The last three properties are linked by the following expression (Slingo, 1989):

$$\tau_c(\lambda) = LWP[a(\lambda) + b(\lambda)/r_e] \qquad (1)$$

where $a(\lambda)$ and $b(\lambda)$ are spectral cloud coefficients.



In the original formulation of OASIM presented in GC2009, $cov$ and $LWP$ data are retrieved from the International Satellite Cloud Climatology Project (ISCCP, from June 1986 to July 2002, and the prior climatology), while $r_e$ is parameterized using

MODIS data normalized with mean values obtained from the literature (please refer to GC2009 for details). Specific modelling of the spectral reflectance of sea foam, affecting the transmittance across the air-sea interface, is also included in OASIM (please refer to Appendix of GC2009).

In addition to the cloud properties necessary for the Slingo model (cloud cover and cloud liquid water path), OASIM requires the following atmospheric input data: surface pressure, $sp$ [mb], wind speed, $ws$ [m/s], relative humidity, $rh$ [%], precipitable

water (absorption by water vapour), $wv$ [cm], and ozone, $oz$ [DU]. Except for ozone, which was obtained from the multiyear dataset of Total Ozone Mapping Spectrometer (TOMS) sensors (from 1979 to May 1993, hereafter referred to as the TOMS climatology), in GC2009, the other four parameters ($sp$, $ws$, $rh$, and $wv$) were acquired from the National Centers for Environmental Prediction (NCEP 1979 – July 2002; Kalnay et al., 1996) reanalysis dataset. In the present implementation, most of these input data are extracted from the ECMWF ERA-Interim dataset (please refer to Section 2.4).

The spatial and temporal resolutions of OASIM are determined by the forcing datasets (i.e., aerosol optical data, cloud property data and atmospheric surface level data). The standard configuration presented in GC2009 is hereby maintained, with a 1-degree horizontal resolution, thereby increasing the temporal frequency to 15 minutes to resolve the diurnal variability and properly compare the output to the temporal resolution of the in situ data considered in the present study. High spatial resolutions and operational-oriented setups are of course possible, provided that forcing data are available: an upgrade in this

direction is under investigation based on the ERA5 dataset recently updated and released (C3S, 2017; Hersbach et al., 2020).

## 2.2 The BGC-Argo float network in the Mediterranean Sea

The BGC-Argo float network represents the first-ever near-real-time (NRT) biogeochemical in situ large-scale ocean observing system (Johnson and Claustre, 2016). The technology is relatively recent, and floats equipped with sensors measuring the main

biogeochemical and optical parameters have now become totally operational (Bittig et al., 2019). A BGC-Argo float operates similarly to an Argo float, collecting vertical profiles from 0-1000 m every 1 to 10 days and transmitting NRT data. The profiles are processed with a variable-specific quality control (QC) approach and are available within 24 hours after data transmission.

The BGC-Argo floats deployed in the Mediterranean Sea since 2012 are equipped with sensors, which, in addition to the

temperature (T), salinity (S) and depth, measure the chlorophyll-a (Chl) fluorescence, downward planar irradiance ($E_d$) at 3 wavelengths (380, 412 and 490 nm) and downward photosynthetically available radiation (DPAR), which represents the integrated amount of the downward planar irradiance in the visible range, i.e., from 400 to 700 nm. Quality-controlled Chl, T and S NRT data are currently available from the Coriolis data centre (Argo Data Management Team, ADMT, manual).

The photosynthetically available radiation (PAR) must be generally derived by spectrally integrating the scalar irradiance (i.e.,

the radiance integrated across the complete solid angle) because phytoplankton utilize light from all directions. In the present





case, BGC-Argo sensors measure the downward irradiance integrated in the 400 to 700 nm range: to avoid confusion, the measured quantity is referred to as DPAR.

QC of radiometric data (DPAR and $E_d$), specifically designed for in situ and remote sensing ocean colour applications, has been implemented by Organelli et al. (2016). The delayed mode (DM) QC approach to identify data corruption by biofouling

and instrument drift based on tests and procedures has been developed by Organelli et al. (2017). In the present work, a BGC-Argo dataset was adopted covering the period between 2012 and 2017 with 3800 profiles (Fig. 2).

Before comparing model values to observations, the irradiance profiles obtained from floats were extrapolated to the surface with a nonlinear fitting procedure only for profiles at a depth shallower than 1.5 metres, with at least 4 measurements in the first 10 metres. In addition, any regions and months containing fewer than 5 profiles were discarded.


## 2.3 The BOUSSOLE mooring buoy

The BOUée pour l'acquiSition d'une Série Optique à Long termE (BOUSSOLE) is a long-term mooring station collecting radiometric and bio-optical properties every 15 min of the topmost 10-m ocean layer (plus a reference, namely, the above-water measurement of the spectral downward planar irradiance) since 2003 (Antoine et al., 2006). It is located in the Ligurian

Sea at 7°54′E and 43°22′N, approximately 32 nautical miles off the French Riviera coast where the water depth is approximately 2440 m (Fig. 2). The measured quantities include the QC-ed multispectral (9 bands) downward planar irradiance above the sea surface and DPAR, covering the period from 2003-2012.

## 2.4 OASIM input data for Mediterranean Sea applications

The input data required by the OASIM model in the present Mediterranean Sea application are shown in Fig. 3, together with the datasets used for the validation (BOUSSOLE and BGC-Argo). In the setup specific for the Mediterranean Sea, developed within the framework of the BIOPTIMOD project, the variables of the cloud and atmospheric properties are extracted from the European Centre for Medium-Range Weather Forecast (ECMWF) ERA-Interim reanalysis dataset (Dee et al., 2011); further details on their pre-processing are provided in Appendix A. The investigated period ranges from 2004 to 2017, with

the input data provided as daily averages and with a spatial resolution of 1 degree.

The use of ERA-Interim to force OASIM is motivated by increasing the coherence between the input cloud properties and surface level atmospheric properties, which, in the GC2009 configuration, were provided by two different datasets (ISCCP and NCEP, respectively). Furthermore, considering that the optical module of the MedBFM model will be upgraded as part of the CMEMS operations, ERA-Interim was chosen since ECMWF products are already operational upstream data for the

Mediterranean Sea regional production centre of the CMEMS (please refer to Salon et al., 2019). The MODIS data for the period from 2000-2017 are derived in the same way as reported in GC2009 but are extended to 2017.





## 2.5 OASIM model validation in the Mediterranean Sea

Data available from the BOUSSOLE mooring station and BGC-Argo floats deployed in the Mediterranean Sea were employed

to validate the OASIM model outputs. To visualize the comparison periods between the in situ data and model outputs, the data availability time windows are shown in Fig. 3. The analysis is performed using datasets from BOUSSOLE (the downward planar irradiance $E_d(0^-)$ at 412.5, 442.5, 490, 510, 555, 560, 665, 670, and 681.25 nm, as well as DPAR) and BGC-Argo floats (the downward planar irradiance $E_d(0^-)$ at 380, 412, and 490 nm and DPAR).

The notation $0^-$ indicates quantities just below the sea surface with any reflections at the air-sea interface removed, and $0^+$ is

applied for the radiances just above the air-sea interface (the light measured before air-sea transmission occurs).

The OASIM outputs for the irradiance are expressed in W m$^{-2}$, while the in situ data for the same parameter are expressed in W cm$^{-2}$ nm$^{-1}$. Therefore, before comparison, the units were standardized in W m$^{-2}$ nm$^{-1}$. Because OASIM simulates data in the range from 350-700 nm centred in 25-nm bins, a linear interpolation was applied to match with measured wavelengths.

Furthermore, the direct ($E_{dir}(0^-)$) and diffuse ($E_{dif}(0^-)$) downward irradiance components simulated by OASIM were

summed to compare them to the measurements of the downward planar irradiance sensors installed on the BOUSSOLE and BGC-Argo floats.

DPAR (µmol quanta m$^{-2}$ s$^{-1}$) was computed from the OASIM output (in standardized units) by integrating the downward planar irradiance as follows:

$$DPAR(0^+) = \frac{10^6}{N_A hc} \int_{400}^{700} [E_{dir}(\lambda, 0^+) + E_{dif}(\lambda, 0^+)]\, \lambda d\lambda \qquad (2)$$

where $N_A$ is the Avogadro number, $h$ is the Planck constant and $c$ is the speed of light. The definition of DPAR in GC2009 relied on integration with a lower bound of 350 nm, while the BOUSSOLE and BGC-Argo float sensors integrate from 400 nm. The model outputs were standardized according to the observations. To compute DPAR($0^-$), which is required for a correct comparison to BGC-Argo float sensors, Eq. (2) was adopted by considering $E_{dir}(0^-)$ and $E_{dif}(0^-)$. Hereafter, $E_d(0^+)$ and $E_d(0^-)$ indicate the sum of the direct and diffuse downward components.


## 3 Results

### 3.1 ERA-Interim and MODIS atmospheric data analysis

A comparison of the ERA-Interim monthly averaged surface level atmospheric variables at grid points corresponding to the

BOUSSOLE mooring buoy and the available observations (sea level pressure, wind speed and relative humidity) by the nearby Cote d'Azur buoy (Météo-France) for the period from 2004-2012 is shown in Fig. 4.





An overall good agreement is observed for the surface pressure between the ECMWF data and in situ observations, while ERA-Interim underestimates the observed relative humidity (D'Ortenzio et al., 2008). The wind magnitude provided by the Cote d'Azur dataset is much larger than that of the ERA-Interim coarse data at 1 degree, as previously reported by Stopa and

Cheung (2014) using data from the U.S. National Data Buoy Center. Large differences occur more frequently during the cold season (from November to April).

In terms of the spatial distribution, the cloud cover follows a clear seasonal cycle (Fig. 5a), with low values during summer in the eastern sub-basins (ION1, ION2, LEV1, LEV2, LEV3, and LEV3) and a high cover during winter in the northern sub-basins (NWM, TYR1, TYR2, ADR1, ADR2, and AEG). The maximum monthly cloud cover during winter reaches

approximately 50%.

However, the aerosol optical thickness exhibits a different spatiotemporal pattern. The highest values are localized in the southwestern Mediterranean (SWM1) between July and August, possibly related to Saharan dust events in the area (Varga et al, 2014). High aerosol thickness values are also found in spring (April and May) in the eastern sub-basins (ION1, ION2, LEV3, and LEV4), which also most likely occurs due to aeolian dust transport (Antoine and Nobileau, 2006).


### 3.2 Validation of the OASIM model at the BOUSSOLE site

The present OASIM configuration produced data with a 15'-temporal resolution at the global scale on a 1-degree mesh. As an example, the simulated daily cycle averaged over March is shown in Fig. 6. As expected, the measured data present a higher variability than that presented by the model, especially when the intra-monthly frequency is considered. The root mean square

difference (RMSD) substantially decreases when considering the monthly average.

The year-by-year RMSD of the OASIM vs. BOUSSOLE relationship remains steady at approximately 0.2 W m$^{-2}$ nm$^{-1}$ (Fig. 7, the red lines). The statistics (Fig. 7, the red dots) indicate a seasonal oscillation: in winter, the high RMSDs and low slopes imply a low prediction skill of the model, whereas in summer, low RMSDs and a slope near 1 indicate a high modelling skill. Furthermore, an averaged day in each month of the time-series [average-day-per-month] was computed, both for the data and

model, discretized in 15'-temporal intervals. The same statistics were applied to the averaged data, and the results reveal a reduction in RMSD (Fig. 7, the cyan dots). The average-day-per-month filtering result indicates that the day-by-day variability is responsible for a substantial and consistent part of the uncertainty (please refer to the averaged values shown in the Fig. 7 panels).

As an additional step, the averaged diurnal cycle grouping all the data in the same month based on the full time series was

considered. For example, all the January data from 2004 to 2012 (9 years) were grouped, and the representative climatological day was derived based on a 15'-temporal resolution. In the case of January, at each wavelength and each 15' interval, there exists a distribution consisting of 31x9 data points (including the data reduction due to sensor failure episodes). In principle, these distributions should be constrained by quite homogeneous conditions, with the same daily zenith angle component, and





similar seasonal conditions, i.e., with all the data grouped by the month. Therefore, assuming that a large number of small
cumulative perturbations affects the variability, the data should be log-normally distributed.

However, the Kolmogorov-Smirnov test revealed that the distributions are not lognormal, neither the BOUSSOLE nor the model data. In fact, the distances between the accumulated empirical distributions and the reference log-normal distribution were almost always larger than the critical distance ($Dcrit=1.36/\sqrt{N_{samples}}$; Bronshtein and Semendyayev, 2013), corresponding to the 5% probability threshold to reject the null hypothesis ($H_0$=the two distributions are the same). Moreover, analysis of the
skewness and excess kurtosis confirmed that the data and model qualitatively exhibit similar distributions: both are negatively skewed, and the tails decay slower than does a Gaussian distribution (images not shown).

### 3.3 Validation of the OASIM model with the BGC Argo float network

The quality-controlled BGC-Argo float dataset adopted in this work contains radiometric measurements acquired from 10:00 to 14:00 local time. To compare OASIM with these BGC-Argo measurements, a point-by-point match-up analysis was performed, where the closest model output to the float measurement at the surface was selected.

The individual match-ups were then spatiotemporally aggregated based on the climatological months and the defined 16 subbasins, as shown in Fig. 2, at each of the three wavelengths and for DPAR separately (Figs. 8 to 11).

At 380 nm, the mean values of the model outputs are overall higher than the observations. Apart from the seasonal cycle, a west-east gradient is also observed, with high values in the Eastern Mediterranean, most likely due to the low cloud cover (Fig. 8ab). Almost all model outputs reveal a positive bias, with the largest differences in the southwestern Mediterranean (SWM2) and southern Adriatic seas (ADR2; up to 0.8 in December), as shown in Fig. 8c. The RMSD exhibits high values in the winter months in the majority of the sub-basins and the lowest values in summer in the Levantine region (LEV1, LEV2, and LEV3;
as shown in Fig. 8d).

Similar to the findings at the BOUSSOLE site, the model attains low skills with respect to the BGC-Argo float observations at 412 nm, with a less pronounced west-east gradient than that indicated by the observations (Fig. 9a,b). Except for the winter months in the southwestern Mediterranean (SWM1), Ionian (ION2) and Levantine (LEV1, LEV2, LEV3) sub-basins, the bias is negative overall (with a bias of up to -0.4, as shown in Fig. 9c). The RMSD does not follow a clear pattern, but it is generally
close to 0.3, with the highest values in December in the Ionian Sea (ION2), as shown in Fig. 9d.

From late spring to autumn, the bias is predominantly negative from the Tyrrhenian Sea eastwards, varying between 0 and -0.2 (Fig. 10c).

At 490 nm, the model values are higher during winter and early spring months, especially in the western sub-basins (Fig. 10a,b). The highest RMSD values are observed in the southwestern Mediterranean (SWM1) in November and December and
in the Levantine region in February (Fig. 10d).

The modelled DPAR values are generally higher than the in situ observations (Fig. 11a,b), with the highest bias and RMSD values observed during the winter period (Fig. 11cd).





### 3.4 Summary of the OASIM model skills in the Mediterranean Sea

The absolute bias of the comparison of the OASIM outputs to the BOUSSOLE data (Tab. 1, upper part) is generally lower than 0.1 W m$^{-2}$ nm$^{-1}$, and the regression slope approaches 1. In particular, the model vs the observations reveals a total negative bias below 20% of the average measured signal, and on average, the RMSD is approximately 30-40%. The regression slopes are lower than 1, indicating a slight underestimation of the model with respect to the observations in terms of the irradiance maxima, particularly at 412 nm (slope=0.66), 670 nm (slope=0.63) and 681.25 nm (slope=0.62).

A lower agreement is observed between OASIM and the BGC-Argo floats (Tab. 1, lower part) than that between OASIM and BOUSSOLE, especially in terms of DPAR. Moreover, in this case, the bias is not always negative, being positive only at the 380-nm wavelength and for DPAR. The correlation for $Ed(\lambda, 0^-)$ indicates a general good agreement between the model and BGC-Argo data (r~0.8), but the slope is below 0.8. Moreover, similar to OASIM vs BOUSSOLE, $Ed(\lambda, 0^-)$ reveals the highest discrepancy at 412 nm, with an RMSD of 0.3 W m$^{-2}$ nm$^{-1}$, a bias of -0.13 W m$^{-2}$ nm$^{-1}$ and a regression slope of 0.51. The comparison to DPAR indicates a positive bias higher than 300 µmol quanta m$^{-2}$ s$^{-1}$, which is high with respect to the BGC-Argo floats than in the BOUSSOLE case.

### 3.5 Comparison of OASIM, BOUSSOLE and BGC-Argo floats in the North-Western Mediterranean Sea

Intercomparison of the model outputs and data obtained from the radiometric sensors of the BOUSSOLE buoy and BGC-Argo floats was possible only when the latter were located in the vicinity of the BOUSSOLE buoy in the NW Mediterranean Sea. Different spatial aggregations of profiles surrounding the fixed buoy were tested, ranging from 1 degree (+/- 0.5 degree from the location of the buoy, as shown in Fig. 12) to the whole northwestern Mediterranean sub-basin (NWM), as shown in Fig. 13. In regard to the 1-degree aggregation, up to 10 BGC-Argo profiles were available per month (Fig. 12, number not shown), while for the sub-basin analysis, the number of available profiles ranged from 8 in October to up to 100 in March (Fig. 13, number not shown).

Monthly climatologies were calculated from 2004-2012 for BOUSSOLE and from 2012-2017 for BGC-Argo, with the model values corresponding to the locations of the instruments limited to the model spatial resolution. The BGC-Argo data were extrapolated to the surface, and the considered profiles followed the required conditions, as described in the previous section. The mean values and standard deviations are therefore shown for 4 different sets of results (Figs. 12 and 13, respectively): the data from the BOUSSOLE buoy (B) and BGC-Argo floats (F) with the corresponding OASIM (BM and FM, respectively) model outputs at 412 and 490 nm (expressed in W m$^{-2}$ nm$^{-1}$) and DPAR (expressed in µmol quanta m$^{-2}$ s$^{-1}$).





A separate comparison of the two different data sources conveyed an overall good agreement, with higher standard deviation
values for the floats (Figs. 12 and 13) than those for the models, revealing a high variability range for the BGC-Argo float
data.

The float values matched with their corresponding OASIM outputs indicated high model outputs at 412 nm, with the largest
differences during the summer months, consistent with Fig. 9, with a positive bias of up to 0.2 W m$^{-2}$ nm$^{-1}$. The highest bias at
the BOUSSOLE site was observed during spring with a similar magnitude (Fig. 13).

The best agreement was generally reached at 490 nm, as is also observed from the NWM column shown in Fig. 10c, especially
during the winter months, where the differences between the two assessed platforms decreased to less than 0.1 W m$^{-2}$ nm$^{-1}$
(Fig. 13). The largest discrepancy was observed during summer, with the float cluster resulting in higher values than those of
the BOUSSOLE buoy (up to 0.2 W m$^{-2}$ nm$^{-1}$, as shown in Fig. 13).

Consistent with the results shown in Fig. 10, major discrepancies arose when comparing DPAR, where the model values
resulted in much higher values than those obtained from the floats, increasing especially during summer (up to 600 µmol
quanta m$^{-2}$ s$^{-1}$ in August, as shown in Fig. 13).

Such inter- and intra-comparisons of the atmospheric radiative transfer model and available radiometric measurement
platforms could also serve as a useful tool to estimate the range of variability when considering optical data from different
sources. The spatial aggregation of measurements to the sub-basin level (i.e., a range of up to 10 degrees) reveals the
preservation of the irradiance seasonal variations. However, much remains to be explained in terms of the sources of both
variabilities, both between the model and float data, especially when considering DPAR, as well as between the two different
sources, such as the floats and fixed buoy.


## 4 Discussion

An extensive comparison of the OASIM model's temporal and spatial variabilities was performed using the BOUSSOLE and
BGC-Argo optical sensor data as references. The results indicated that, in general, the model reproduced the variability in the
spectral downward irradiance in the Mediterranean Sea, which depends on the spatiotemporal scale. In particular, considering
the OASIM applications within biogeochemical models equipped with a multispectral in-water optical module, the impact will
be different according to the specific scale under investigation. In terms of fine temporal scales, it was observed that a large
part of the discrepancy expressed by the RMSD occurred due to the day-to-day variability, similar to the findings reported by
Somayajula et al. (2018). When this temporal scale is filtered out, the RMSD decreases by half or more. Consistently, the
assessed uncertainty (an RMSD of approximately 10%, BIAS lower than 10%, and slope > 85%) should be considered in
multiannual simulations. In fact, a key parameter such as the primary productivity (strongly affected by the irradiance) exhibits

a dominant component of the variance at the seasonal scale (Lazzari et al., 2012), while the day-to-day variability and inter-annual variability seem to be less important (Di Biagio et al. 2019). In contrast, for short-term forecasts (i.e., 10 days; Salon et al., 2019), the day-to-day variability in the downward irradiance could be relevant. Clearly, to fully investigate the high-frequency RMSD variability, simulations should be refined by increasing the spatial resolution of the model in the region surrounding the BOUSSOLE site (e.g., C3S, 2017; Hersbach et al., 2020).

Apart from the diurnal cycle, the cloud cover and aerosols are major drivers in modulating the variability in the downward irradiance, with aerosols being subordinate to the cloud cover. These two input parameters show a high seasonal variability (Figs. 14 and 15, respectively). In the Mediterranean area, the cloud cover is high during fall and winter, while the aerosol optical thickness is high during spring and summer (Fig. 5). The winter maximum of the cloud cover corresponds to the maximum RMSD at all wavelengths considered (Fig. 14), while the minimum RMSD occurs in July when the cloud cover is also at its annual minimum. A consistent variability is obtained by computing the regression slope (Figs. 14 and 15). These results are in line with a previous multimodel analysis (Somayajula et al., 2018) indicating that models present a negative bias when the cloud cover is higher than 70%. Such an underestimation could impact phytoplankton dynamics modelling, and thus, further analysis should be conducted. The multimodel comparison by Nielsen et al. (2014) revealed that the Slingo liquid-optics model tends to overestimate the cloud optical thickness and therefore underestimates the irradiance.

The presence of a systematic underestimation, following the mechanism described in Nielsen et al. (2014), likely affects all temporal scales. However, in the present study, it was shown that the RMSD and regression slope are greatly improved when filtering out the day-to-day variability. This indicates that high-resolution sampling in OASIM could notably improve the model results.

The cloud cover spatial distribution indicates that similar conclusions are obtained based on the model vs BGC-Argo skills with the results obtained at the BOUSSOLE site, namely, at least for $E_d(\lambda = 490, 0^-)$ and $E_d(\lambda = 380, 0^-)$, the increase in the match between the model and data is modulated by the cloud cover (Figs. 5a, 8, 9, and 10). In contrast, $E_d(\lambda = 412, 0^-)$ seems to be less influenced by clouds with the exception of the minimum RMSD during summer in the eastern area.

Apart from cloud dynamics, aerosols play a role in landlocked regions such as the Mediterranean Sea (Papadimas et al., 2008; Nabat et al., 2015). The AOT reveals a variability at least an order of magnitude larger than that of the asymmetry parameter or even more than that of the single scattering albedo. The minimum RMSD observed in July does not correspond to an extreme AOT value, as observed in the case of the cloud cover. The AOT generally decreases with increasing wavelength (Fig. 15). However, a reduced model skill is observed at 412 and 682.25 nm. The MODIS aerosol data are extrapolated from the 7 wavelength channels of the satellite sensors (i.e., 470, 550 and 660 nm in the visible region), which could possibly explain the high uncertainty at 412 nm. In terms of the spatial heterogeneity, comparing the model and BGC-Argo floats (Figs. 5b and 8, 9, and 10, respectively), the role of aerosols in the modulation of the model bias and RMSD does not appear to have a clear interpretation.




The low skill at a specific wavelength could also be explained in terms of the wavelength discretization in the OASIM model: the current model spectral resolution of 25 nm could be refined near 412 and 682.25 nm to investigate whether this would reduce the BIAS.


In addition to the seasonal indicators discussed above, the interannual trends of $E_d(\lambda, 0^-)$ and DPAR were investigated both for the measured data and model results. Given the strong seasonal cycle present in all the properties considered, low-pass filters (i.e., moving averages) were applied to the data before the regression.

The results for both datasets were biased by the gaps in the acquisitions, which introduced spurious trends. Therefore, the focus
was on the model inputs instead (ECMWF products and MODIS aerosol data) and the outputs corresponding to a 14-year gap-free time series both in spatial and temporal terms. The model outputs averaged over the Mediterranean basin indicated a low interannual variability, or at least the procedures adopted based on the low-pass filtering and subsequent regression could not identify clear trends (image not shown). Moreover, the analysis performed demonstrated that the cloud cover interannual variability spans a 2% range and the aerosol optical thickness at 490 nm exhibits an approximately 10% variability with a
maximum in 2009. $E_d(\lambda = 490, 0^-)$ and DPAR reveal an even lower interannual variability of approximately 1%.

**5 Conclusions**

The BOUSSOLE mooring station provides a dataset with a high temporal resolution at a fixed point, while the BGC-Argo floats at least allow the partial resolution of the spatial variability (certain regions contain no floats), although with a lower spectral resolution than that of BOUSSOLE. The OASIM model is useful for the integration of all this information. The results indicate an overall good agreement between the model outputs and in situ references, highlighting a clear seasonal variability
in the model capabilities, with a generally high performance during summer.

The analysis conveys that an important source of the discrepancy between the model and data is the intra-monthly variability, which can be ascribed to cloud cover dynamics (the RMSD is reduced with a low cloud cover). Both the seasonal and intra-monthly variabilities in the model skill may be related to cloud property data, e.g., their coarse resolution.

In regard to climatological simulations, the high skill in terms of the monthly averaged irradiance is probably sufficient to
properly constrain biogeochemical dynamics, and attention should be paid in the case of short-term simulations, when biogeochemical processes such as chlorophyll acclimation exhibit the same time scale as the relatively large fluctuations observed in the RMSD.



In this case, in addition to improved cloud parameterizations, the use of daily resolved aerosol data could possibly reduce the model uncertainties, for example, the EUMETSAT polar multisensor aerosol optical property product (PMAP), available since

2014, or other products provided by the Copernicus Atmosphere Monitoring Services (CAMS).

Moreover, novel atmospheric models with improved mathematical descriptions of cloud dynamics and advanced numerical solvers may better simulate both clear-sky and cloudy-sky components (Hogan and Bozzo, 2018).

Among the wavelengths considered for the downward planar irradiance, 412 and 681 nm appear to result in large uncertainties. However, at this stage, it is not possible to ascertain the reasons for such different skills, and further investigations are required.




## Appendix A

In the present application in the Mediterranean Sea, certain input variables are not directly available from the ERA-Interim dataset in the form required by OASIM, so further processing is needed, as detailed in Table A1.

*Table A1. Correspondence between the input variables required in OASIM (left column) and the implementation in the Mediterranean Sea (right column), with specific pre-processing steps.*

| **Cloud properties** (file modcld*.dat read by the OASIM rdatopt.F subroutine) | |
|---|---|
| *OASIM variable* | *Present implementation* |
| cloud cover [%] | ERA-Interim total cloud cover – tcc [(0-1)] multiplied by 100 |
| cloud optical thickness [-] | computed within OASIM, Eq. (1) |
| cloud liquid water path [g m$^{-2}$] | ERA-Interim total column cloud liquid water – tclw [kg m$^{-2}$] multiplied by 1000 |
| cloud droplet effective radius [µm] | based on MODIS climatology, as in GC2009 |
| **Aerosol properties** (file modaer*.dat read by the OASIM rdatopt.F subroutine) | |
| *OASIM variable* | *Present implementation* |
| aerosol optical thickness [-] | based on MODIS data, as in GC2009 |
| aerosol asymmetry parameter [-] | based on MODIS data, as in GC2009 |
| aerosol single scattering albedo [-] | based on MODIS data, as in GC2009 |
| **Atmospheric properties** (file opt*.dat read by the OASIM rdatopt.F subroutine) | |
| *OASIM variable* | *Present implementation* |
| surface pressure [mb] | ERA-Interim surface pressure - sp [Pa] divided by 100 |
| wind speed [m/s] | ERA-Interim 10-metre wind speed [m/s] |





| relative humidity [%] | following ECMWF ERA-Interim[*], the relative humidity [%] is computed as the ratio between the water vapour pressure (which depends on the 2-metre dewpoint temperature, d2 m [K]) and saturation water vapour pressure (which depends on the 2-metre temperature, t2 m [K]), multiplied by 100<br>*https://www.ecmwf.int/en/faq/do-era-datasets-contain-parameters-near-surface-humidity |
|---|---|
| ozone [DU] | ERA-Interim total column ozone - tco3 [kg m$^{-2}$] divided by a factor of 2.1414 x 10$^{-5}$ |
| precipitable water [cm] | following Gregg and Carder (1990), the precipitable water [cm] is computed as a function of the saturated water vapour pressure, surface pressure and sea level atmospheric pressure (Garrison and Adler, 1990) |




**Code availability**

The OASIM Fortran code is publicly downloadable, together with the reference input data, from http://gmao.gsfc.nasa.gov/research/oceanbiology

**Data availability**

ECMWF ERA-Interim:

https://www.ecmwf.int/en/forecasts/datasets/archive-datasets/reanalysis-datasets/era-interim

BOUSSOLE mooring buoy:

http://www.obs-vlfr.fr/Boussole/html/project/boussole.php

Cote d'Azur mooring buoy:

https://mistrals.sedoo.fr/HyMeX/

BGC-Argo:

https://www.ocean-ops.org/board?t=argo

http://www.coriolis.eu.org/Data-Products

http://www.argodatamgt.org/

**Author contribution**

PL and SS designed the study and the numerical experiments and PL carried them out supported by WWG. PL and ET performed the model data comparison. VV was in charge of the BOUSSOLE mooring buoy maintenance and performed the quality-control of the optical data. DA provided data and funding through the BOUSSOLE project. EO compiled the BGC-Argo database and performed the quality-control of the optical data. FDO provided funding and contributed with floats

deployments. All authors contributed to the interpretation of the results. PL prepared the manuscript with contributions from all co-authors.

**Competing interests**

The authors declare that they have no conflict of interest.


**Acknowledgements**:

This work was performed within the framework of the BIOPTIMOD CMEMS Service Evolution project. CMEMS is implemented by Mercator Ocean International within the framework of a delegation agreement with the European Union.

The authors acknowledge Météo-France for supplying the data and the HyMeX database teams (ESPRI/IPSL and

SEDOO/Observatoire Midi-Pyrénées) for their help in accessing the data.





BOUSSOLE is funded by the European Space Agency (ESA), contract 4000119096/17/I-BG, and by the Centre National d'Etudes Spatiales (CNES) with support of the French Oceanographic Fleet (https://doi.org/10.18142/1).

This study was supported by the following research projects funding BGC-Argo floats: NAOS (funded by the Agence Nationale de la Recherche in the frame of the French 'Equipement d'avenir' program, grant agreement ANR J11R107-F), remOcean (funded by the European Research Council, grant agreement 246777), Argo-Italy (funded by the Italian Ministry of Education, University and Research), and the French Bio-Argo program (Bio-Argo France; funded by CNES-TOSCA, LEFE Cyber, and GMMC).

We acknowledge the ECMWF for the ERA-Interim dataset (source: www.ecmwf.int). These data are published under a Creative Commons Attribution 4.0 International (CC BY 4.0), https://creativecommons.org/licenses/by/4.0/. ECMWF does not accept any liability whatsoever for any error or omission in the data, their availability, or for any loss or damage arising from their use.

Authors thank Valentina Mosetti (OGS) for figures editing.



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





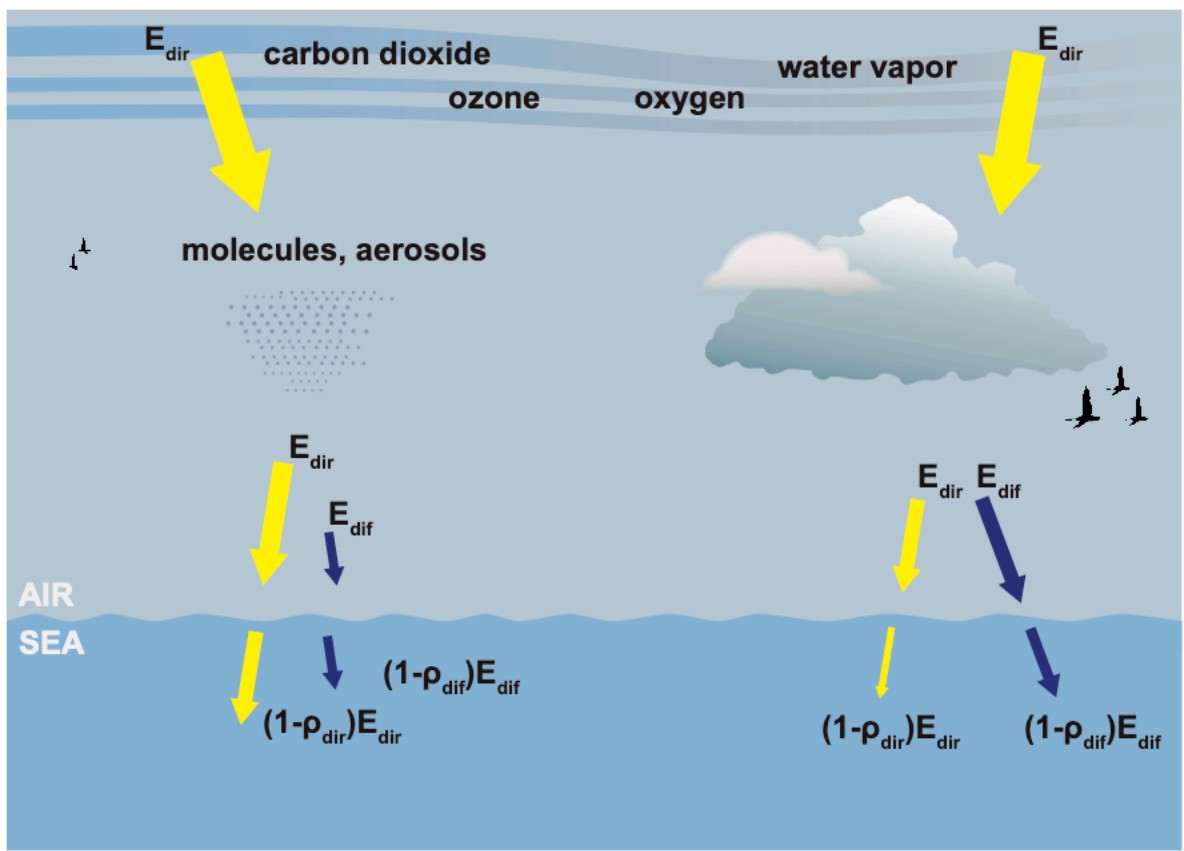

**Figure 1. Irradiance pathways in OASIM under clear skies (left) and cloudy skies (right): $E_{dir}$ is the direct downwelling irradiance, $E_{dif}$ is the diffuse downwelling irradiance, and $\rho_{dir}$ and $\rho_{dif}$ are the direct and diffuse surface reflectances, respectively (the size of the**
**arrows approximates the relative contributions; the figure is modified from Fig. 1 of Gregg and Rousseaux, 2017).**

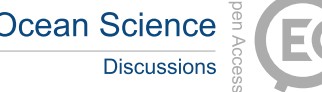

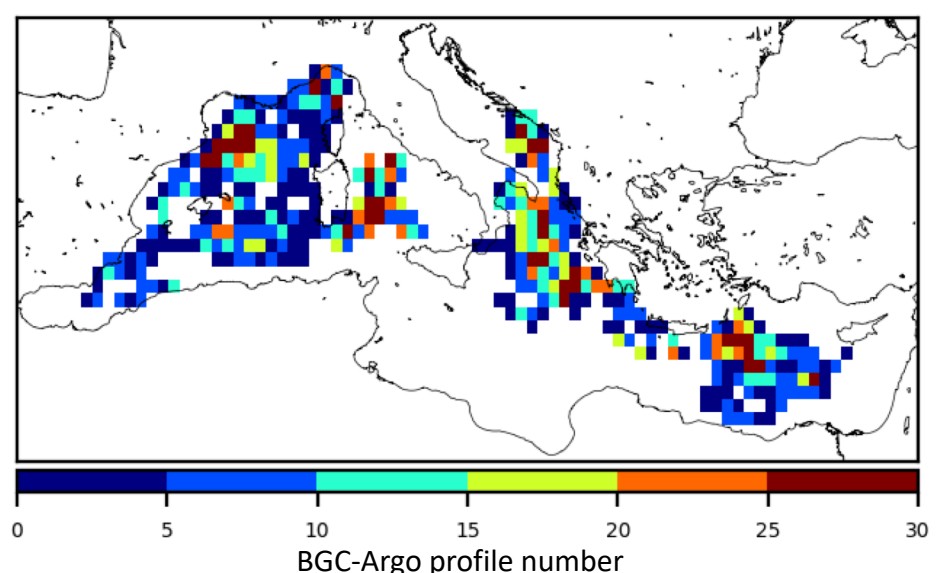

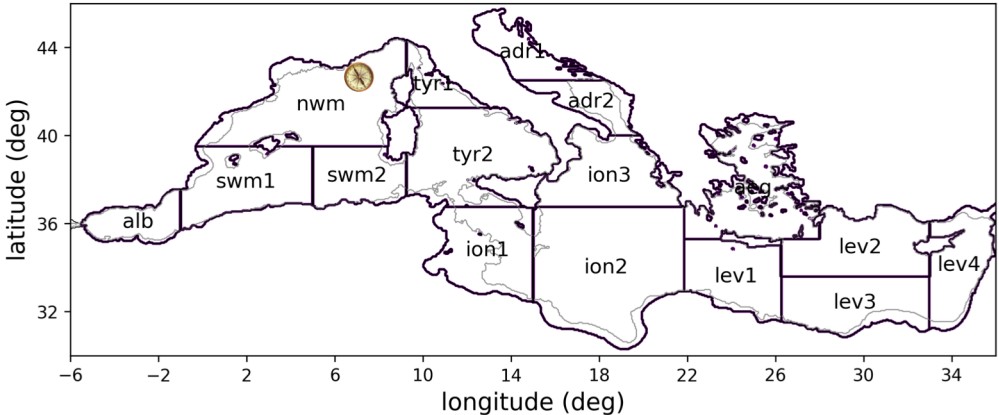

**Figure 2. Top panel: density of the BGC-Argo float profiles in a 0.5°x0.5° window covering the period from 2012-2017. Bottom panel: sub-basin division and corresponding area short names as defined in Salon et al. (2019). The compass indicates the**
**BOUSSOLE buoy site (7°54'E, 43°22'N).**



| | 2000 | 2001 | 2002 | 2003 | 2004 | 2005 | 2006 | 2007 | 2008 | 2009 | 2010 | 2011 | 2012 | 2013 | 2014 | 2015 | 2016 | 2017 |
|---|---|---|---|---|---|---|---|---|---|---|---|---|---|---|---|---|---|---|
| **OASIM input** | | | | | | | | | | | | | | | | | | |
| cloud properties | | | | | | | | ECMWF ERA INTERIM | | | | | | | | | | |
| aerosol properties | | | | | | | | MODIS | | | | | | | | | | |
| atmospheric optics | | | | | | | | ECMWF ERA INTERIM | | | | | | | | | | |
| **OASIM output** | | | | | | | | | | | | | | | | | | |
| $E_{dir}$, $E_{dif}$ | | | | | | | | OASIM | | | | | | | | | | |
| **Observations** | | | | | | | | | | | | | | | | | | |
| **BOUSSOLE** | | | | | | BOUSSOLE | | | | | | | | | | | | |
| **BGC-Argo** | | | | | | | | | | | | | BGC-ARGO | | | | | |

**Figure 3. Availability of the model input (cloud, aerosol, and atmospheric data) and radiometric output data ($E_{dir}$ and $E_{dif}$) and corresponding observational datasets for the validation in the present work. The acronym definitions are reported in Sections 2.1,**
**2.2 and 2.3.**



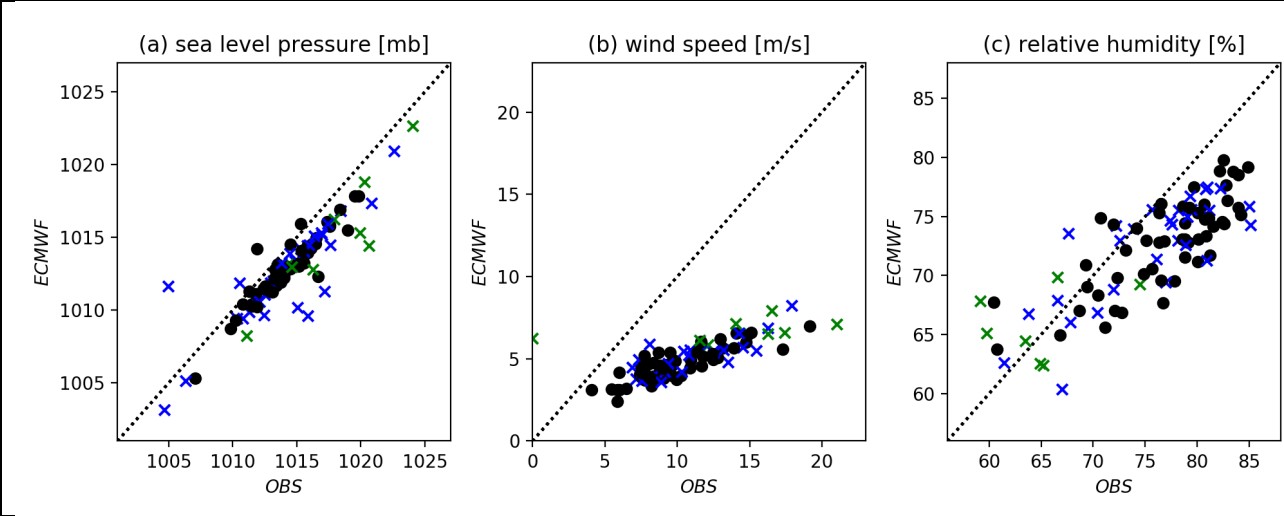

**Figure 4. Comparison of the ECMWF ERA-Interim monthly averaged data of the sea level pressure (a), wind speed (b) and relative humidity (c) and corresponding averages computed from the observations measured by the Cote d'Azur buoy near the BOUSSOLE platform. The symbols represent the different months: January to April (blue crosses), May to October (black dots), and November to December (green crosses).**



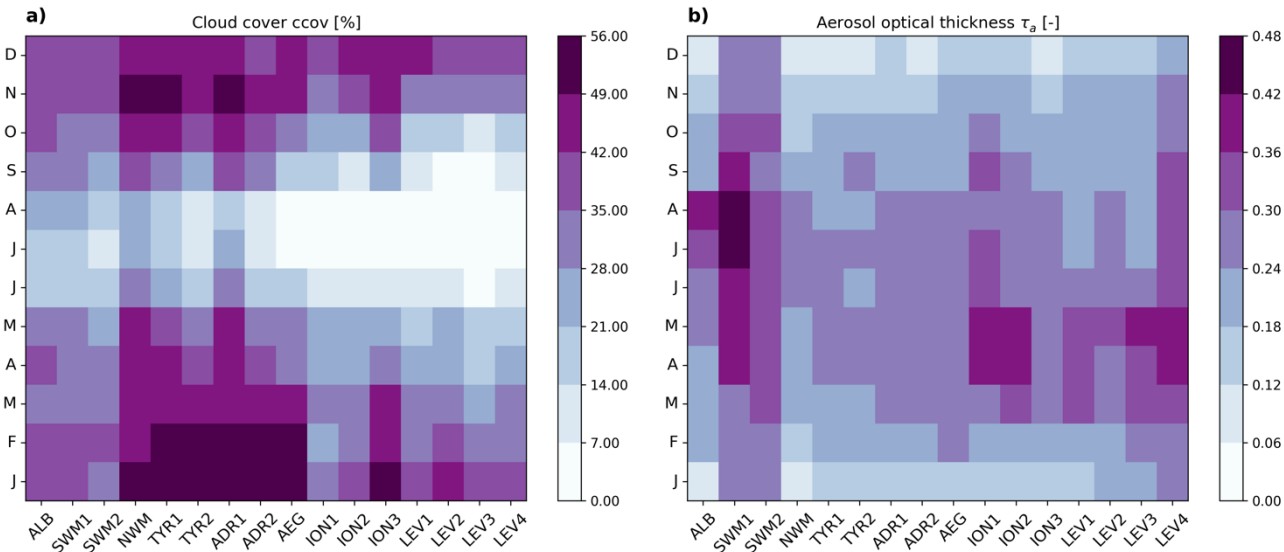

**Figure 5.** ERA-Interim data of the cloud cover (panel a) and MODIS aerosol optical thickness at 475 nm (panel b), aggregated spatiotemporally based on the climatological months (y-axis) and in the 16 subbasins (x-axis), as shown in Fig. 2.





**Figure 6. Multispectral downward planar irradiance ($E_d(\lambda, 0^-)$) simulated by OASIM (blue lines) and measured at BOUSSOLE (red lines). The wavelengths considered are those measured by the BOUSSOLE sensors for the average March data derived from the time series. For each panel, the reported statistics (RMSD, Bias, r, and regression slope) are related both to the high-frequency signal (with a temporal resolution of 15'; top left) and to the average day in the considered month (top right). The vertical bars indicate the variance in the monthly averaged values of the average day.**






**Figure 7. RMSD (left column) and slope (right column) of the relationship between the OASIM and BOUSSOLE downward planar irradiance values ($E_d(\lambda, 0^-)$) at the nine wavelengths ($\lambda$, in nm) and DPAR. The vertical bars in the left panels of each column are the BOUSSOLE (grey) and model (green) annual averages. The RMSD (W m$^{-2}$ nm$^{-1}$) and the slope (dimensionless) are marked in red and black, respectively, with the lines indicating the annual averages and the dots indicating the monthly averages. The monthly statistics are further separated by filtering out the day-by-day variability and are shown as the cyan dots for the RMSD and fuchsia dots for the slope. The averages over the time series are annotated in the panels.**






**Figure 8. Results of the match-up between the model (M) and BGC-Argo float observations (O) for $E_d(\lambda = 380, 0^-)$, aggregated spatiotemporally based on the climatological months and 16 subbasins. The top figures show the mean values of the model (M, panel a) and observations (O, panel b). BIAS and RMSD are shown in panels c and d, respectively. The unit is W m$^{-2}$ nm$^{-1}$, and BIAS and RMSD are dimensionless due to the normalization by the average values. The grey colour represents the points in space**
**and time for which fewer than 5 match-ups were available.**





**Figure 9.** Same as Fig. 8 for $E_d(\lambda = 412, 0^-)$.







Figure 10. Same as Fig. 8 for $\mathbf{E_d}(\lambda = 490, 0^-)$.





**Figure 11. Same as Fig. 8 for DPAR; in this case, the MEAN values are expressed in μmol quanta m⁻² s⁻¹.**






**Table 1. Summary of the model skill compared to the available data from the BOUSSOLE buoy (from 2004 to 2012) and BGC-Argo floats (from 2012 to 2017) for the irradiance (Ed) at the different wavelengths (WL) and for DPAR. RMSD, bias, and Y-int are expressed in W m$^{-2}$ nm$^{-1}$, and in regard to DPAR, the same statistical indicators are expressed in µmol quanta m$^{-2}$ s$^{-1}$, while all the other indicators (regression r and slope) are dimensionless, where N is the number of match-ups between the model and observations. For the BOUSSOLE comparison, the green numbers are derived by filtering out the day-to-day variability (i.e., the intra-monthly variability). Given the large number of samples, all statistics are significant (p-value < 0.05). For the RMSD and BIAS, the percentage values normalized by average data are reported in parentheses.**

| | BOUSSOLE vs OASIM-ECMWF [2004-2012] | | | | | |
|---|---|---|---|---|---|---|
| **WL** | **RMSD** | **BIAS** | **R** | **SLOPE** | **Y-int** | **N** |
| **412.5** | 0.14 (34.1%) | -0.05 (-11.4%) | 0.83 | 0.66 | 0.08 | 55207 |
| | 0.04 (10.3%) | -0.05 (-11.4%) | 0.99 | 0.88 | 0.00 | |
| **442.5** | 0.17 (33.7%) | -0.01 (-1.2%) | 0.84 | 0.77 | 0.09 | 110952 |
| | 0.04 (7.5%) | -0.01 (-1.3%) | 0.99 | 1.00 | -0.01 | |
| **490** | 0.19 (34.4%) | -0.01 (-2.0%) | 0.84 | 0.76 | 0.10 | 112138 |
| | 0.04 (7.8%) | -0.01 (-2.1%) | 0.99 | 1.00 | -0.02 | |
| **510** | 0.20 (34.6%) | -0.02 (-3.9%) | 0.83 | 0.74 | 0.10 | 112013 |
| | 0.04 (7.6%) | -0.02 (-4.0%) | 0.99 | 0.98 | -0.02 | |
| **555** | 0.19 (33.6%) | 0.02 (3.4%) | 0.85 | 0.83 | 0.10 | 55231 |
| | 0.05 (9.1%) | 0.02 (3.4%) | 0.99 | 1.05 | -0.03 | |
| **560** | 0.19 (35.6%) | 0.00 (0.3%) | 0.83 | 0.76 | 0.11 | 106532 |
| | 0.04 (8.4%) | 0.00 (0.3%) | 0.99 | 1.02 | -0.02 | |
| **665** | 0.17 (34.2%) | -0.02 (-4.7%) | 0.84 | 0.75 | 0.09 | 76165 |
| | 0.04 (7.5%) | -0.02 (-4.8%) | 0.99 | 0.99 | -0.03 | |
| **670** | 0.17 (39.8%) | -0.05 (-12.0%) | 0.79 | 0.63 | 0.08 | 32688 |
| | 0.04 (10.6%) | -0.05 (-12.2%) | 0.98 | 0.92 | -0.02 | |
| **681.25** | 0.16 (36.3%) | -0.08 (-18.3%) | 0.81 | 0.62 | 0.07 | 110286 |
| | 0.04 (10.3%) | -0.08 (-18.4%) | 0.99 | 0.85 | -0.02 | |
| **DPAR$_{400,700}$** | 249.15 (34.7%) | -12.63 (-1.8%) | 0.84 | 0.74 | 146.35 | 106012 |
| | 50.71 (7.3%) | -12.59 (-1.8%) | 0.99 | 0.98 | -9.27 | |
| | BGC-Argo floats vs OASIM-ECMWF [2012-2017] | | | | | |
| **WL** | **RMSD** | **BIAS** | **R** | **SLOPE** | **Y-int** | **N** |
| **380** | 0.15 (31%) | 0.07 (14%) | 0.81 | 0.69 | 0.22 | 2624 |
| **412** | 0.3 (33%) | -0.13 (-15%) | 0.81 | 0.51 | 0.31 | 2569 |
| **490** | 0.29 (26%) | -0.01 (-1%) | 0.82 | 0.64 | 0.39 | 2258 |
| **DPAR$_{400,700}$** | 452.77 (48%) | 327.91 (35%) | 0.74 | 0.82 | 492.55 | 1042 |






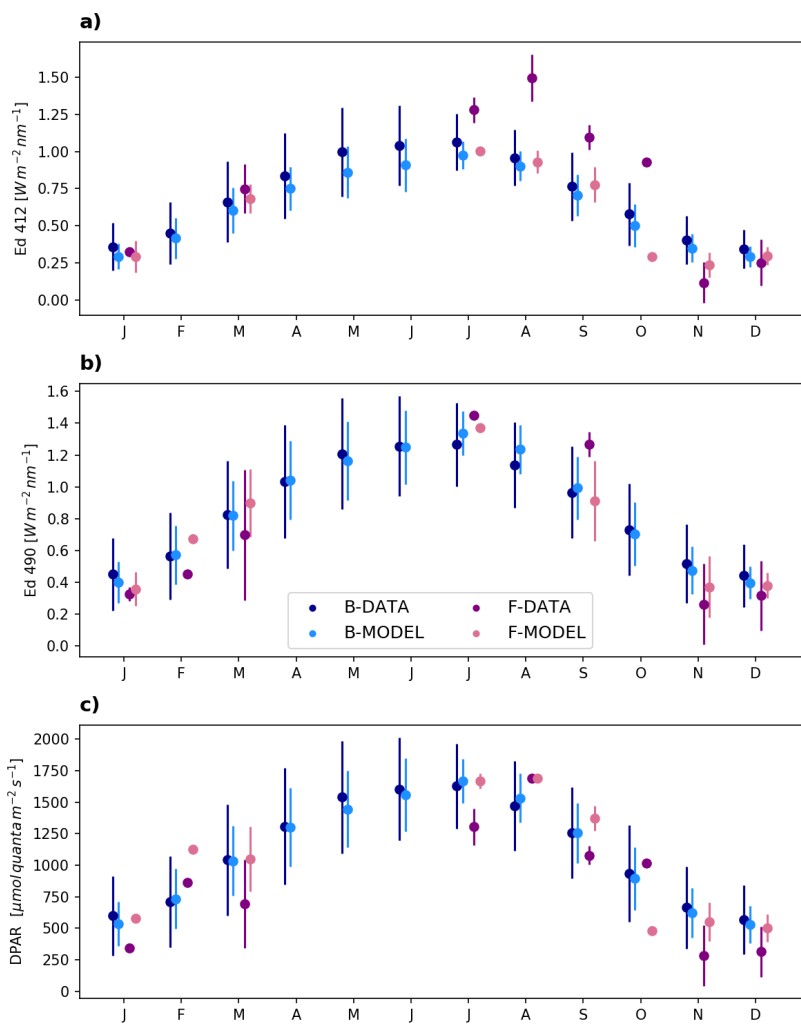

**Figure 12. Scatter plots of the climatological monthly mean values and standard deviations of the downward irradiance $E(\lambda = 412, 490, 0^-)$ (W m$^{-2}$ nm$^{-1}$) and DPAR (µmol quanta m$^{-2}$s$^{-1}$). The blue dots represent the mean values of the measurements at the BOUSSOLE site and the corresponding model outputs, whereas the lilac points display the BGC-Argo values and model means, resulting from the match-up of all available profiles within the 1-degree window (+/- 0.5 degree N/S and W/E from the location of the BOUSSOLE buoy). Note that the points corresponding to each month are horizontally shifted to increase the readability.**





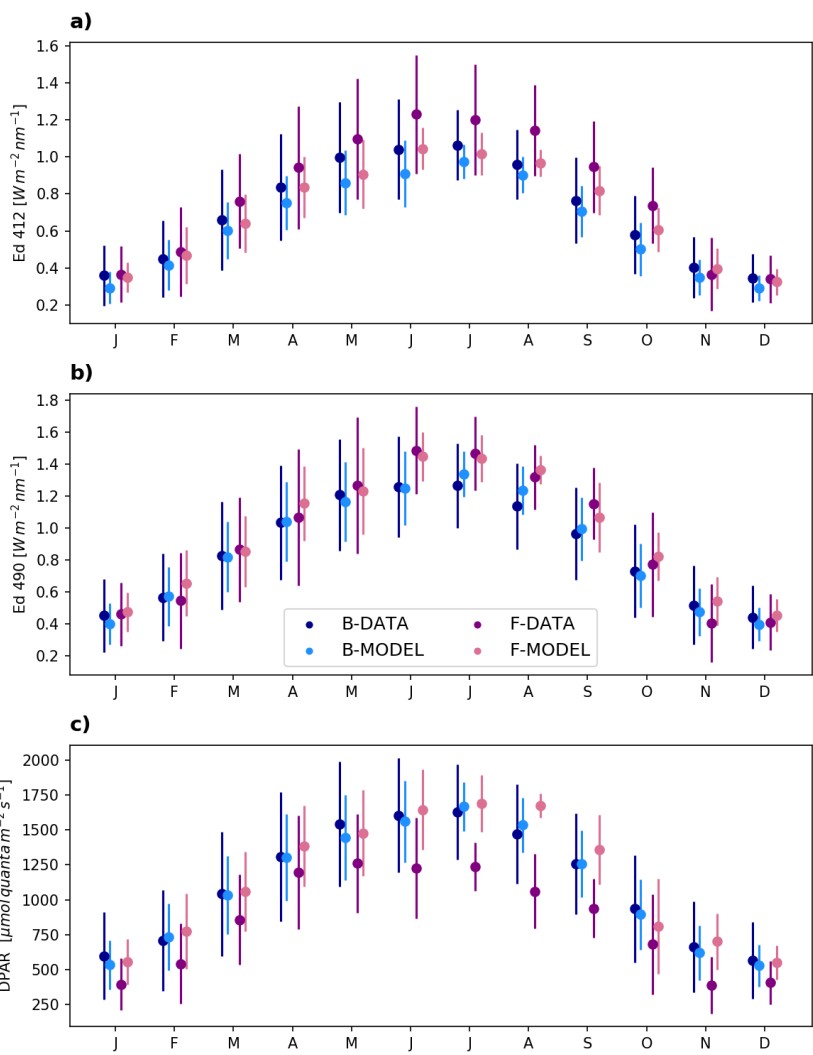

**Figure 13.** Scatter plots of the climatological monthly mean values and standard deviations of the downward irradiance $E_{0-}(\lambda =$ 412, 490 nm) (W m$^{-2}$ nm$^{-1}$) and DPAR (µmol quanta m$^{-2}$s$^{-1}$). The blue dots represent the mean values of the measurements at the BOUSSOLE site and the corresponding model outputs, whereas the lilac points display the BGC-Argo values and model means, resulting from the match-up of all available profiles in the northwestern Mediterranean (NWM) sub-basin. Note that the points corresponding to each month are horizontally shifted to increase the readability.




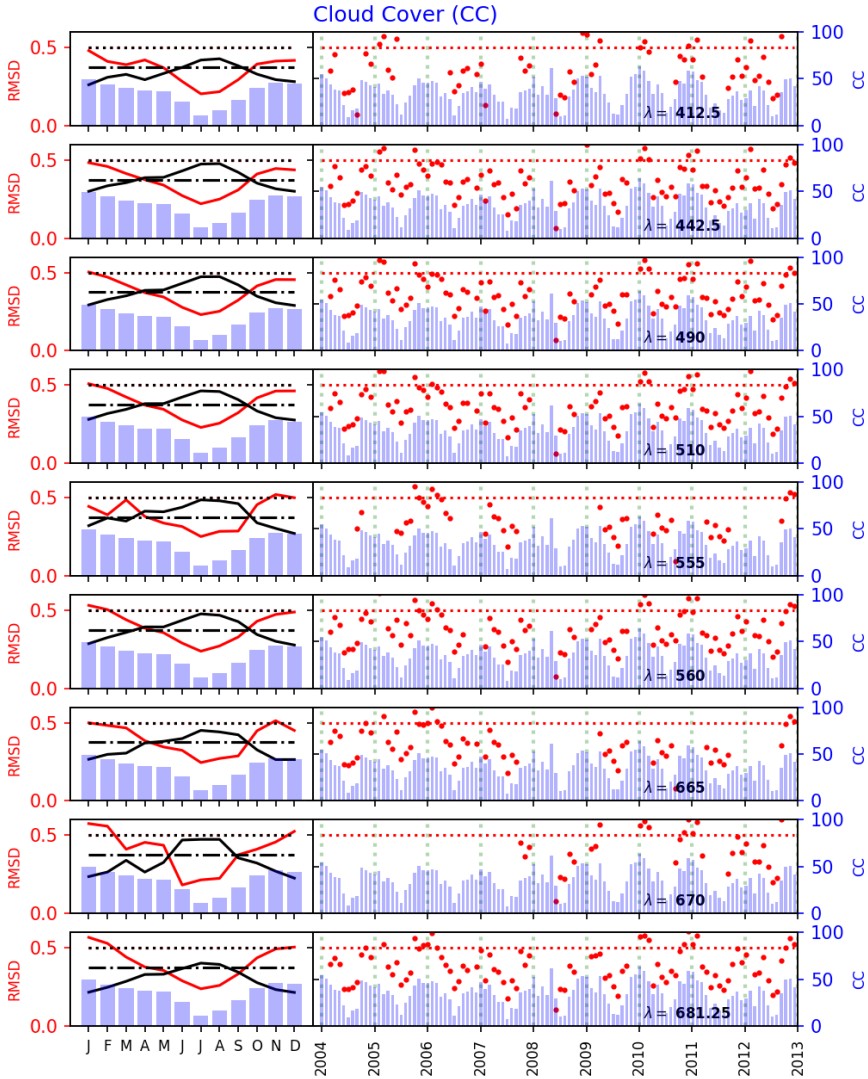

**Figure 14. Comparison of the OASIM downward irradiance ($E_d(\lambda, 0^-)$ at the nine wavelengths) to the BOUSSOLE data from 2004 to 2012 in terms of the RMSD and regression slope and their relationship with the ECMWF ERA-Interim cloud cover (CC). The left section of each panel shows the monthly climatology of the RMSD (normalized by its averaged value; red lines and labels) and regression slope (normalized by its averaged value; black line), superimposed on the monthly climatology of the cloud cover (in %, blue bars and labels). Regression slope thresholds at 1 (dotted black line) and 0.75 (dot-dashed black line) are also shown. The right section of each panel shows the monthly means of the time series of the RMSD (red dots, with a 0.5 value; red dotted line), superimposed on the monthly means of the time series of the cloud cover (blue bars and labels).**





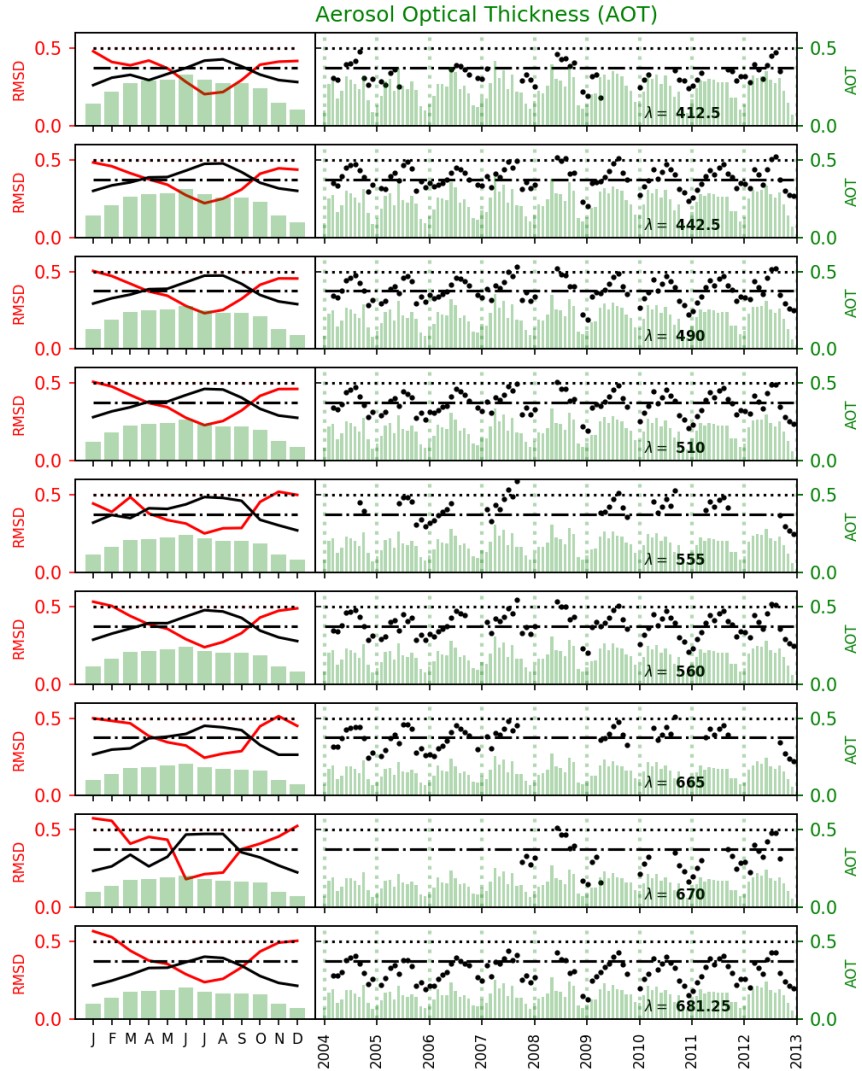

**Figure 15. Comparison of the OASIM downward irradiance ($E_d(\lambda, 0^-)$ at the nine wavelengths) to the BOUSSOLE data from 2004 to 2012 in terms of the RMSD and regression slope and their relationship with the MODIS aerosol optical thickness. The left section of each panel shows the monthly climatology of the RMSD (normalized by its averaged value, red lines and labels) and regression slope (normalized by its averaged value, black line), superimposed on the monthly climatology of the aerosol optical thickness (green bars and labels). Regression slope thresholds at 1 (dotted black line) and 0.75 (dot-dashed black line) are also shown. The right section of each panel shows the monthly means of the time series of the regression slope (black dots, with the 1 and 0.75 thresholds shown; dotted and dot-dashed black lines, respectively), superimposed on the monthly means of the time series of the aerosol optical thickness (green bars and labels).**