# Peer review of "Assessment of the spectral downward irradiance at the surface of the Mediterranean Sea using the OASIM ocean-atmosphere radiative model"

_Ocean Science, 2020_

## Referee Comment (RC1) · Anonymous Referee #1 · 20 Dec 2020

The work entitled "Assessment of the spectral downward irradiance at the surface of the Mediterranean Sea using the OASIM ocean-atmosphere radiative model" by Lazzari et al., 2020 assessed the surface spectral downward irradiance over the Mediterranean Sea using OASIM ocean-atmosphere radiative model with high temporal resolution BOUSSOLE buoy data and BGC-Argo data. The article presented the spatiotemporal analysis of the downward planar irradiance at the ocean-atmosphere interface.

This work emphasizes the need of a good quality controlled in situ data such as from BOUSSOLE buoy and growing network of BGC-Argo floats data in model evaluations.

[Figure]

Availability of such data is highly relevant in addressing both the climatological as well as day-to-day impacts of light variability on ocean biology.

This work will be a very good contribution towards utilizing and the significance of high resolution data (both spatial and temporal), towards data assimilation into biogeochemical models. In my view this work definitely paves a way in considering the aspects of spatial and temporal variability considering the model resolutions and how they can be improved in future. Specifically, towards the role of light input to the models. The methodology and the representation of the data were substantially given in explaining the scientific concepts.

The proposed scientific approach and the methods applied are very well represented by the authors. The explanation of the results, discussion and conclusions are not exhaustive and very appropriately given in a more concise manner in relation to the model design in accordance with both the in situ data sets. All the explanations of results and discussion were well referenced emphasizing the role of different parameters in towards the model errors and biases. The quality of the figures, and their explanations were very much appropriate, clear and concise.

I think the manuscript would be considered for publication after making the following small corrections.

Specific comment:

Comment 1: I suggest the addition of a table explaining the abbreviations used in the article (different models, model parameters etc.,). Even though having explained them in the text looks fine, but still having a Table is highly appreciated.

Minor corrections:

P1Line 21: Table 1 shows that except for 670 nm for BOUSSOLE buoy, and DPAR values 0.79 for buoy and 0.71 for BGC-Argo, the correlation values (R) are higher than 0.8 and with removing the day-to-day they are higher than 0.9. This should be

mentioned in the abstract.

Please correct the correlation r as R.

P5Line 141: correct QC-ed as QC-Ed

(comment, no need to response) Figure 4. shows that the wind speeds are very much underestimated compared to ECMWF. It can be seen that the wind speeds go as high as 20 m/s, and a high variability is observed. Considering OSAIM model at the ocean-atmosphere interface, what possible impact does this have on model simulations? I just wanted to know.
* * *

---

## Referee Comment (RC2) · Anonymous Referee #2 · 30 Dec 2020

Reviewer comments on

Assessment of the spectral downward irradiance at the surface of the Mediterranean Sea using the OASIM ocean-atmosphere radiative model by

Paolo Lazzari 1 , Stefano Salon 1 , Elena Terzić 1 , Watson W. Gregg 2 , Fabrizio D'Ortenzio 3 , Vincenzo Vellucci 4 , Emanuele Organelli 3,5 and David Antoine 6,3

The manuscript presents a model-observation validation study that compares the OASIM spectral radiation transfer model results at the surface of Mediterranean Sea

to BOUSSOLE and BGC-Argo observations. The comparison has been done in the framework of Copernicus Marine Environment Monitoring Service in order to prepare for development of a new multispectral bio-optical model with advanced data assimilation for marine research. The atmospheric input to OASIM is taken from the ECMWF ERA Interim reanalysis data, and the comparisons cover the years 2004-2017 (which is not clearly indicated). The general result is that at monthly level the model and observations correspond each other well, but daily variations are large, depending mostly on cloud dynamics.

Large amounts of valuable observation data has been used and model simulations performed. The radiative transfer model and observations are described in a sufficient extent, the results extensively presented and analysed. For validation, suitable statistical methods seem to be used. Data and model availability is documented as required. However, the reader is somewhat lost within this vast material as the presentation is not focused and clear enough. Sometimes it is even difficult to understand what the authors want to say. The motivation and aims of this study should be stated clearly in the introduction and the conclusions tied to them. You might consider if all the figures are really necessary in order to support your conclusions.

For example, in the abstract (L10-20) you write that "observations are combined with model outputs to analyse the spatial and temporal variabilities in the downward planar irradiance at the ocean-atmosphere interface". In fact you validate a radiation transfer model against ocean observations. In the introduction, you refer to development possibilities, including everything from advanced assimilation of satellite observations to improved coupled biogeochemical models applying bio-optical in-water light propagation algorithms. How exactly the comparison of a classical atmospheric radiative transfer model to the (new) in-situ observations, which is the topic of the present study, will contribute to those developments, is not detailed or prioritized.

I have one major question for you to consider, as you use ERA reanalysis data and work within Copernicus monitoring services. ERA5 includes output of spectrally integrated surface downward UV (0.20-0.44 $\mu$m) and SW (0.20 - 12 $\mu$m) radiation fluxes (https://cds.climate.copernicus.eu/cdsapp#!/dataset/reanalysis-era5-single-levels?tab=overview). ERA interim did not yet contain the separate UV flux. These fluxes result from application of the ECMWF radiation transfer model that is fully integrated in the atmospheric model at each time step. The previous version before the scheme described by Hogan and Bozzo, 2018 (your reference at L386-387) was applied for ERA interim and ERA5.

I would suggest you to compare the ERA5 UV and SW to the BOUSSOLE and Argo DPAR and integrated UV measurements. This would give you a possibility to understand if these variables of operational or reanalysis NWP models were sufficient as input to MedBFM within CMEMS. It would give you a basis to request CAMS, ECMWF future output of some more spectral details of the downward global, direct and clear-sky surface radiation fluxes, e.g. UV, visible, near-IR, or more, separately. If this would succeed, you would not need at all the coupling of OASIM within CMEMS but would benefit from the integrated advanced radiation scheme within the ECMWF model?

You have shown that the aerosols play a minor role compared to clouds (L324). In any case, use of detailed aerosol information in the radiation schemes of the NWP models would make the additional use of atmospheric MODIS aerosol data unnecessary for the coupled ocean bio-optical modules. You can find out about CAMS and ECMWF treatement of aerosol information in Bozzo et. al., https://doi.org/10.5194/gmd-13-1007-2020, 2020 and references therein. Some regional NWP models plan to use CAMS aerosol in near-real-time (e.g. Rontu et al, 2020, https://doi.org/10.3390/atmos11020205) for the weather forecast (but with spectrally simple radiation schemes, with output not sufficient for your purposes).

A few specific minor comments follow:

L31-34. Please clarify the complicated sentence, what does it mean?

L121 and elsewhere: photosynthetically available radiation -> photosynthetically *active* radiation

L133-134. How to understand this: at a depth shallower than 1.5 metres, with at least 4 measurements in the first 10 metres ?

Section 3.1. What is the role of surface pressure and wind from the point of view of (solar) radiation flux comparison?

L.296 and elsewhere. How to compare fluxes in different units, 600 $\mu$mol quanta m -2 s -1 v.s. W m -2 nm -1 ?

L331-336. You are effectively saying that for the radiation flux results it is more important that clouds are in correct place in correct time than how the details of liquid cloud optics are treated in the simulated clouds. Which is true, of course.

L383-387. See the general comments.

---

## Referee Comment (RC3) · M. Baird (Referee) · 18 Jan 2021

This paper quantifies the skill of the OASIM radiative transfer model in the Mediterranean Sea region. Although there is no (cited) comparison of the skill of other atmospheric optical models, the OASIM performance is impressive, and would be both good news and useful information for the groups using, or considering using, OASIM. Another interesting feature of the manuscript is the use of the Bio-Argo array for model assessment. As this will no doubt become the primary means of assessing large-scale marine atmospheric models in the future, the technique developed in the manuscript

will be interesting to others.

While there are lots of excellent figures, the manuscript itself is quite short and does not make as much of the analysis as it could – leaving some room for additions, which would help to make the manuscript more useful and citable. A couple of ideas for strengthening the manuscript:

1. The OASIM is assess primarily based on Ed(0-), the downwelling planar irradiance. It is mentioned that the BOUSSOLE observations have above water observations. Analysis of skill of Ed(0+) would help to distinguish between OASIM errors in the atmosphere and in the transmission across the sea-surface interface. This distinction would be useful for both improving OASIM, and also for those using OASIM.

2. Following on from 1., the article distinguishes between errors in cloud-free and cloudy days. I wonder whether a somewhat similar, but more diagnostically-useful distinction might be the fractions of direct and diffuse radiation. As per point 1 this might be more useful for OASIM developers and also for those attempting to apply these results outside the Mediterranean where lower sun angles might change the balance of direct vs diffuse for the same cloud.

3. The extrapolation of the Bio-Argo data to the surface sounds like a critical step. Can you give more details? For example, do you assume an exponential decay with depth?

Minor comments. 1. L23 "daily variability" – this is somewhat ambiguous. Do you mean between days or within days?

2. L24 replace 'high' with a number.

3. 'cloud dynamics and seasonality" – the former is a process, the later a timescale. They don't quite fit together within one phrase.

4. L30 remove "notably"

5. L47 -51. Confusing. Tried to say too much in one sentence.

6. L83 'while' – this is not a 'while'

7. L84 – remove 'the' before 'aerosol'.

8. L107 "resolve the diurnal variability" – this is within a day right?

9. L108 'properly' is a subjective, rather than objective, adjective.

10. L115 remove 'totally'

11. L134. At this point I didn't know how you were defining regions.

12. L166. Would it be more accurate to say "W m-2 per waveband"?

13. L167 "to W m-2"

14. L202 – 15' and 1-degree – most people would write 15 minutes and 1o!

15. L257 – stick with bias = model – observation. Don't say a negative bias, but a bias of -20%. Otherwise it can get confusing.

16. L264, L265 – subscript of d different to elsewhere.

17. L294 'float cluster" is new wording that is unnecessary.

18. L354. More details on the wavelength discretisation. For example, Does 412 sit in a 400 to 425 nm band?

19. L373 "The OASIM model . . . this information" – Conclusions need tighter sentence than this.

20. L386-387 – this manuscript could be more helpful for motivating this sentiment as per main point 2.

21. Fig 4. Wind speed at what height?

22. Fig 12-Fig 13 – It took me a moment to work out what 'B' and 'F' meant, especially since you them both BioArgos and floats. In figure BOUSSOLE and BioArgo would be

quicker for interpretation.

---

## Author Comment (AC1) · 18 Feb 2021

We thank Reviewer #1 for the useful comments. The point by point response is provided below in blue font and the proposed text modifications in red.

**Anonymous Referee #1**

The work entitled "Assessment of the spectral downward irradiance at the surface of the Mediterranean Sea using the OASIM ocean-atmosphere radiative model" by Lazzari et al., 2020 assessed the surface spectral downward irradiance over the Mediterranean Sea using OASIM ocean-atmosphere radiative model with high temporal resolution BOUSSOLE buoy data and BGC-Argo data. The article presented the spatiotemporal analysis of the downward planar irradiance at the ocean-atmosphere interface. This work emphasizes the need of a good quality controlled in situ data such as from BOUSSOLE buoy and growing network of BGC-Argo floats data in model evaluations. Availability of such data is highly relevant in addressing both the climatological as well as day-to-day impacts of light variability on ocean biology. This work will be a very good contribution towards utilizing and the significance of high resolution data (both spatial and temporal), towards data assimilation into biogeochemical models. In my view this work definitely paves a way in considering the aspects of spatial and temporal variability considering the model resolutions and how they can be improved in future. Specifically, towards the role of light input to the models. The methodology and the representation of the data were substantially given in explaining the scientific concepts. The proposed scientific approach and the methods applied are very well represented by the authors. The explanation of the results, discussion and conclusions are not exhaustive and very appropriately given in a more concise manner in relation to the model design in accordance with both the in situ data sets. All the explanations of results and discussion were well referenced emphasizing the role of different parameters in towards the model errors and biases. The quality of the figures, and their explanations were very much appropriate, clear and concise. I think the manuscript would be considered for publication after making the following small corrections.

Thanks for the encouraging comments.

Specific comment:

Comment 1: I suggest the addition of a table explaining the abbreviations used in the article (different models, model parameters etc.,). Even though having explained them in the text looks fine, but still having a Table is highly appreciated.

We agree with the reviewer, we propose to add a table in the Appendix with the definitions of the abbreviations, an example is shown below.

| Abbreviation | Long name |
| --- | --- |
| OASIM | Ocean–Atmosphere Spectral Irradiance Model |
| BOUSSOLE | BOUée pour l'acquiSition d'une Série Optique à Long termE |
| BGC-Argo float | Biogeochemical Argo float |
| OCR-VC | Ocean Colour Radiometry Virtual Constellation |
| MedBFM | Mediterranean Sea biogeochemical operational model system within CMEMS |
| CMEMS | Copernicus Marine Environment Monitoring Service |
| ECMWF | European Centre for Medium-Range Weather Forecasts |
| ERA-Interim | ECMWF reanalysis |
| … | |

Minor corrections:

1. P1Line 21: Table 1 shows that except for 670 nm for BOUSSOLE buoy, and DPAR values 0.79 for buoy and 0.71 for BGC-Argo, the correlation values (R) are higher than 0.8 and with removing the day-to-day they are higher than 0.9. This should be mentioned in the abstract.

   Thanks for the comment we propose to add this text in the Abstract.

   The correlations (R) between the data and model are always higher than 0.6. With the exception of DPAR and the 670 nm channel, correlation values are always higher than 0.8 and, when removing the inter-daily variability, they are higher than 0.9.

2. Please correct the correlation r as R.

Thanks, we will correct.

3. P5Line 141: correct QC-ed as QC-Ed (comment, no need to response)

4. Figure 4. shows that the wind speeds are very much underestimated compared to ECMWF. It can be seen that the wind speeds go as high as 20 m/s, and a high variability is observed. Considering OSAIM model at the ocean atmosphere interface, what possible impact does this have on model simulations? I just wanted to know.

Sensitivity test to meteorological inputs were performed by Gregg and Carder (1990), showing that pressure and mean wind speed produced differences in surface spectral irradiance less than 1% in terms of RMS model error over the 350-700 nm range, much less than air-mass type, visibility and total ozone. More specifically, their Fig. 5 shows that the ratio Ed(0-)/Ed(0+) mainly remains larger than 0.90 for wind speed ranging 0-15 m/s and two visibility values (5 and 25 km). The ratio decreases to 0.85 only for visibility equal to 25 km, absence of wind and solar zenith angle around 80 degree.

Furthermore, according to Gregg and Carder (1990), direct and diffuse sea surface reflectance can be decomposed in specular and sea foam-dependent reflectance. Foam reflectance is affected by sea-surface roughness, which in turn has previously been related to wind stress and, secondarily, to wind speed (Koepke, 1984). We will add this information in the methods Section.

[Figure]

Figure R1. Multispectral downward planar irradiance **Ed($\lambda$,0+)** simulated by OASIM (blue lines) and measured at BOUSSOLE (red lines). The wavelengths considered are those measured by the BOUSSOLE sensors for the average March data derived from the time series. For each panel, the reported statistics (RMSD, Bias, r, and regression slope) are related both to the high-frequency signal (with a temporal resolution of 15'; top left) and to the average day in the considered month (top right). The vertical bars indicate the variance in the monthly averaged values of the average day.

In order to estimate the impact of surface pressure and wind in the model-observation comparison, we show in Fig. R1 the multispectral downward planar irradiance (**Ed($\lambda$,0+)**) simulated by OASIM (blue lines) and measured at BOUSSOLE (red lines)

for the month of March. Comparing Fig. R1 with Fig. 6 proposed in the submitted version for **Ed(λ,0-)**, we show that differences, related to the atmospheric parameters have in general a low impact on the results, otherwise we would observe a much higher deterioration in model performance when computing **Ed(λ,0-)** from **Ed(λ,0+)**.

In order to provide a general overview of the impact of the parameters indicated by the Reviewer, we extracted Ed(λ,0+) from model and from BOUSSOLE and reported the skill in Tab. R1 in analogy to Tab. 1 proposed in the manuscript. The differences in the skill (e.g. percentual RMSD and BIAS) between computation of **Ed(λ,0-)** from **Ed(λ,0+)** indicates that the model is in general slightly better in computing **Ed(λ,0+)** than **Ed(λ,0-)** but the differences are, in any case of second order, so is the impact of surface pressure and wind. RMSD is only marginally affected with <1% differences. The BIAS for **Ed(λ,0+)** shows <5% differences.

Table R1. Summary of the model skill compared to the available data from the BOUSSOLE buoy (from 2004 to 2012) and BGC-Argo floats (from 2012 to 2017) for the irradiance (**Ed(λ,0+)**) at the different wavelengths (WL) and for DPAR. RMSD, bias, and Y-int are expressed in W m-2 nm-1, while all the other indicators (regression R and slope) are dimensionless, where N is the number of match-ups between the model and observations. For the BOUSSOLE comparison, the green numbers are derived by filtering out the day-to-day variability (i.e., the intra-monthly variability). Given the large number of samples, all statistics are significant (p-value < 0.05). For the RMSD and BIAS, the percentage values normalized by average data are reported in parentheses.

| BOUSSOLE vs OASIM-ECMWF [2004-2012] | | | | | | |
|---|---|---|---|---|---|---|
| WL | RMSD | BIAS | R | SLOPE | Y-int | N |
| 412.5 | 0.15 (34.1%) | -0.04 (-9.5%) | 0.83 | 0.66 | 0.08 | 55239 |
| | 0.04 (10.0%) | -0.04 (-9.6%) | 0.99 | 0.88 | 0.00 | |
| 442.5 | 0.18 (33.6%) | 0.00 (0.6%) | 0.84 | 0.77 | 0.09 | 111010 |
| | 0.04 (7.2%) | 0.00 (0.5%) | 0.99 | 1.00 | -0.01 | |
| 490 | 0.20 (34.4%) | 0.00 (-0.1%) | 0.84 | 0.76 | 0.10 | 112186 |
| | 0.04 (7.4%) | 0.00 (-0.2%) | 0.99 | 1.00 | -0.02 | |
| 510 | 0.20 (34.6%) | -0.01 (-2.0%) | 0.83 | 0.74 | 0.10 | 112071 |
| | 0.04 (7.2%) | -0.01 (-2.1%) | 0.99 | 0.98 | -0.02 | |
| 555 | 0.20 (33.4%) | 0.03 (5.1%) | 0.85 | 0.83 | 0.10 | 55309 |
| | 0.05 (8.6%) | 0.03 (5.0%) | 0.99 | 1.05 | -0.03 | |
| 560 | 0.20 (35.5%) | 0.01 (2.3%) | 0.83 | 0.76 | 0.11 | 106660 |
| | 0.04 (7.9%) | 0.01 (2.3%) | 0.99 | 1.02 | -0.02 | |
| 665 | 0.18 (34.1%) | -0.02 (-3.0%) | 0.84 | 0.75 | 0.09 | 76247 |
| | 0.04 (7.1%) | -0.02 (-3.1%) | 0.99 | 0.99 | -0.03 | |
| 670 | 0.17 (39.6%) | -0.04 (-9.3%) | 0.79 | 0.63 | 0.08 | 32733 |
| | 0.04 (10.1%) | -0.04 (-9.5%) | 0.98 | 0.92 | -0.02 | |
| 681.25 | 0.17 (36.4%) | -0.08 (-16.4%) | 0.81 | 0.62 | 0.07 | 110418 |
| | 0.05 (10.3%) | -0.07 (-16.6%) | 0.99 | 0.85 | -0.02 | |

---

## Author Comment (AC2) · 18 Feb 2021

We thank Reviewer #2 for the useful comments. The point by point response is provided below in blue font and the proposed text modifications in red.

**Anonymous Referee #2**

Reviewer comments on Assessment of the spectral downward irradiance at the surface of the Mediterranean Sea using the OASIM ocean-atmosphere radiative model by Paolo Lazzari 1 , Stefano Salon 1 , Elena Terzic 1 , Watson W. Gregg 2 , Fabrizio ´D'Ortenzio 3 , Vincenzo Vellucci 4 , Emanuele Organelli 3,5 and David Antoine 6,3 The manuscript presents a model-observation validation study that compares the OASIM spectral radiation transfer model results at the surface of Mediterranean Sea to BOUSSOLE and BGC-Argo observations. The comparison has been done in the framework of Copernicus Marine Environment Monitoring Service in order to prepare for development of a new multispectral bio-optical model with advanced data assimilation for marine research. The atmospheric input to OASIM is taken from the ECMWF ERA Interim reanalysis data, and the comparisons cover the years 2004-2017 (which is not clearly indicated). The general result is that at monthly level the model and observations correspond each other well, but daily variations are large, depending mostly on cloud dynamics.

     Thanks, we propose to explicitly mention the time span of the analysis that is 2004-2017 both in the abstract and Introduction.

Large amounts of valuable observation data has been used and model simulations performed. The radiative transfer model and observations are described in a sufficient extent, the results extensively presented and analysed. For validation, suitable statistical methods seem to be used. Data and model availability is documented as required. However, the reader is somewhat lost within this vast material as the presentation is not focused and clear enough. Sometimes it is even difficult to understand what the authors want to say. The motivation and aims of this study should be stated clearly in the introduction and the conclusions tied to them.

     We tried to better explain the motivations specifically introducing the applications in the abstract and in the introduction the application regarded to biogeochemistry we propose the following corrections

"This activity has been carried out within the framework of the CMEMS Service Evolution project BIOPTIMOD, which is aimed at the development of a new multispectral bio-optical model that will include the integration of MedBFM with data provided by both BGC-Argo floats and multispectral satellite sensors (e.g., the Ocean and Land Colour Instrument, OLCI, on-board Sentinel3-A and Sentinel3-B; Donlon et al, 2012)."

You might consider if all the figures are really necessary in order to support your conclusions.

     Thanks for the comments, but we think that all the figures are interesting and useful to support our findings.

For example, in the abstract (L10-20) you write that "observations are combined with model outputs to analyse the spatial and temporal variabilities in the downward planar irradiance at the ocean-atmosphere interface". In fact you validate a radiation transfer model against ocean observations. In the introduction, you refer to development possibilities, including everything from advanced assimilation of satellite observations to improved coupled biogeochemical models applying bio-optical in-water light propagation algorithms. How exactly the comparison of a classical atmospheric radiative transfer model to the (new) in-situ observations, which is the topic of the present study, will contribute to those developments, is not detailed or prioritized.

     The atmospheric multispectral input data are necessary to resolve the multispectral propagation of light along the water column. Evaluating the uncertainties of these input data is fundamental for all

the future developments concerning multispectral bio-optical models. Since OASIM is used by our group and other groups we think that this assessment can be useful and interesting for the scientific community.

I have one major question for you to consider, as you use ERA reanalysis data and work within Copernicus monitoring services. ERA5 includes output of spec trally integrated surface downward UV (0.20-0.44 µm) and SW (0.20 - 12 µm) radiation fluxes (https://cds.climate.copernicus.eu/cdsapp#!/dataset/reanalysis-era5-singlelevels?tab=overview). ERA interim did not yet contain the separate UV flux. These fluxes result from application of the ECMWF radiation transfer model that is fully integrated in the atmospheric model at each time step. The previous version before the scheme described by Hogan and Bozzo, 2018 (your reference at L386-387) was applied for ERA interim and ERA5. I would suggest you to compare the ERA5 UV and SW to the BOUSSOLE and Argo DPAR and integrated UV measurements. This would give you a possibility to understand if these variables of operational or reanalysis NWP models were sufficient as input to MedBFM within CMEMS. It would give you a basis to request CAMS, ECMWF future output of some more spectral details of the downward global, direct and clearsky surface radiation fluxes, e.g. UV, visible, near-IR, or more, separately. If this would succeed, you would not need at all the coupling of OASIM within CMEMS but would benefit from the integrated advanced radiation scheme within the ECMWF model?

Concerning the comparison of the ERA5 model with the instruments presented in this manuscript we remark that neither the BOUSSOLE buoy nor the BGC-Argo floats optical sensors measure  UV (0.20-0.44 µm) and SW (0.20 - 12 µm) radiation fluxes and a comparison with the ERA5 data is not straightforward.
In any case, as mentioned in the manuscript, we plan to further explore the optical output provided by CAMS and/or ECMWF in future studies and applications.

You have shown that the aerosols play a minor role compared to clouds (L324). In any case, use of detailed aerosol information in the radiation schemes of the NWP models would make the additional use of atmospheric MODIS aerosol data unnecessary for the coupled ocean bio-optical modules. You can find out about CAMS and ECMWF treatement of aerosol information in Bozzo et. al., https://doi.org/10.5194/gmd-13-1007-2020, 2020 and references therein. Some regional NWP models plan to use CAMS aerosol in near-real-time (e.g. Rontu et al, 2020, https://doi.org/10.3390/atmos11020205) for the weather forecast (but with spectrally simple radiation schemes, with output not sufficient for your purposes).

Thanks for the comment, we propose to add these references in the revised manuscript.

A few specific minor comments follow:

1. L31-34. Please clarify the complicated sentence, what does it mean?

Thanks, we propose to rephrase with the following sentence:
Such data may be exploited to improve the calibration and tuning of the bio-optical models embedded in three-dimensional global and regional physical-biogeochemical coupled models. Radiometric data availability will further increase with the development of new autonomous profiling floats dedicated to ocean colour measurements (Leymarie et al., 2018) and with the expanding data streams provided by the Ocean Colour Radiometry Virtual Constellation (OCR-VC) satellite.

2. L121 and elsewhere: photosynthetically available radiation -> photosynthetically *active* radiation

Ok, we will substitute with photosynthetically active radiation.

3. L133-134. How to understand this: at a depth shallower than 1.5 metres, with at least 4 measurements in the first 10 metres ?

We updated the text proposing the following information:

Before comparing model values to observations, the irradiance profiles obtained from floats were extrapolated to the surface with an exponential fitting procedure based on the *curve_fit* tool of the python package scipy. Further, we required profiles to have at least one measurement in the first 1.5 m depth from sea surface and to have at least 4 measurements in the first 10 m. In addition, any sub-basin (as defined in Fig.2) and month containing fewer than 5 profiles was discarded.

4. Section 3.1. What is the role of surface pressure and wind from the point of view of (solar) radiation flux comparison?

We copy the same response given to Reviewer#1 [item 4] since the comments are similar.

Sensitivity test to meteorological inputs were performed by Gregg and Carder (1990), showing that pressure and mean wind speed produced differences in surface spectral irradiance less than 1% in terms of RMS model error over the 350-700 nm range, much less than air-mass type, visibility and total ozone. More specifically, their Fig. 5 shows that the ratio Ed(0-)/Ed(0+) mainly remains larger than 0.90 for wind speed ranging 0-15 m/s and two visibility values (5 and 25 km). The ratio decreases to 0.85 only for visibility equal to 25 km, absence of wind and solar zenith angle around 80 degree.

Furthermore, according to Gregg and Carder (1990), direct and diffuse sea surface reflectance can be decomposed in specular and sea foam-dependent reflectance. Foam reflectance is affected by sea-surface roughness, which in turn has previously been related to wind stress and, secondarily, to wind speed (Koepke, 1984). We propose to add this information in the methods Section.

[Figure]

Figure R1. Multispectral downward planar irradiance **Ed(λ,0+)** simulated by OASIM (blue lines) and measured at BOUSSOLE (red lines). The wavelengths considered are those measured by the BOUSSOLE sensors for the average March data derived from the time series. For each panel, the reported statistics (RMSD, Bias, r, and regression slope) are related both to the high-frequency signal (with a temporal resolution of 15'; top left) and to the average day in the considered month (top right). The vertical bars indicate the variance in the monthly averaged values of the average day.

In order to estimate the impact of surface pressure and wind in the model-observation comparison, we show in Fig. R1 the multispectral downward planar irradiance

(**Ed(λ,0+**)) simulated by OASIM (blue lines) and measured at BOUSSOLE (red lines) for the month of March. Comparing Fig. R1 with Fig. 6 proposed in the submitted version for **Ed(λ,0-)**, we show that differences, related to the atmospheric parameters have in general a low impact on the results, otherwise we would observe a much higher deterioration in model performance when computing **Ed(λ,0-)** from **Ed(λ,0+)**.

In order to provide a general overview of the impact of the parameters indicated by the Reviewer, we extracted Ed(**λ,0+**) from model and from BOUSSOLE and reported the skill in Tab. R1 in analogy to Tab. 1 proposed in the manuscript. The differences in the skill (e.g. percentual RMSD and BIAS) between computation of **Ed(λ,0-)** from **Ed(λ,0+)** indicates that the model is in general slightly better in computing **Ed(λ,0+)** than **Ed(λ,0-)** but the differences are, in any case of second order, so is the impact of surface pressure and wind. RMSD is only marginally affected with <1% differences. The BIAS for **Ed(λ,0+)** shows <5% differences.

Table R1. Summary of the model skill compared to the available data from the BOUSSOLE buoy (from 2004 to 2012) and BGC-Argo floats (from 2012 to 2017) for the irradiance (**Ed(λ,0+)**) at the different wavelengths (WL) and for DPAR. RMSD, bias, and Y-int are expressed in W m-2 nm-1, while all the other indicators (regression R and slope) are dimensionless, where N is the number of match-ups between the model and observations. For the BOUSSOLE comparison, the green numbers are derived by filtering out the day-to-day variability (i.e., the intra-monthly variability). Given the large number of samples, all statistics are significant (p-value < 0.05). For the RMSD and BIAS, the percentage values normalized by average data are reported in parentheses.

| BOUSSOLE vs OASIM-ECMWF [2004-2012] | | | | | | |
|---|---|---|---|---|---|---|
| WL | RMSD | BIAS | R | SLOPE | Y-int | N |
| 412.5 | 0.15 (34.1%) | -0.04 (-9.5%) | 0.83 | 0.66 | 0.08 | 55239 |
| | 0.04 (10.0%) | -0.04 (-9.6%) | 0.99 | 0.88 | 0.00 | |
| 442.5 | 0.18 (33.6%) | 0.00 (0.6%) | 0.84 | 0.77 | 0.09 | 111010 |
| | 0.04 (7.2%) | 0.00 (0.5%) | 0.99 | 1.00 | -0.01 | |
| 490 | 0.20 (34.4%) | 0.00 (-0.1%) | 0.84 | 0.76 | 0.10 | 112186 |
| | 0.04 (7.4%) | 0.00 (-0.2%) | 0.99 | 1.00 | -0.02 | |
| 510 | 0.20 (34.6%) | -0.01 (-2.0%) | 0.83 | 0.74 | 0.10 | 112071 |
| | 0.04 (7.2%) | -0.01 (-2.1%) | 0.99 | 0.98 | -0.02 | |
| 555 | 0.20 (33.4%) | 0.03 (5.1%) | 0.85 | 0.83 | 0.10 | 55309 |
| | 0.05 (8.6%) | 0.03 (5.0%) | 0.99 | 1.05 | -0.03 | |
| 560 | 0.20 (35.5%) | 0.01 (2.3%) | 0.83 | 0.76 | 0.11 | 106660 |
| | 0.04 (7.9%) | 0.01 (2.3%) | 0.99 | 1.02 | -0.02 | |
| 665 | 0.18 (34.1%) | -0.02 (-3.0%) | 0.84 | 0.75 | 0.09 | 76247 |
| | 0.04 (7.1%) | -0.02 (-3.1%) | 0.99 | 0.99 | -0.03 | |
| 670 | 0.17 (39.6%) | -0.04 (-9.3%) | 0.79 | 0.63 | 0.08 | 32733 |
| | 0.04 (10.1%) | -0.04 (-9.5%) | 0.98 | 0.92 | -0.02 | |
| 681.25 | 0.17 (36.4%) | -0.08 (-16.4%) | 0.81 | 0.62 | 0.07 | 110418 |
| | 0.05 (10.3%) | -0.07 (-16.6%) | 0.99 | 0.85 | -0.02 | |

5. L.296 and elsewhere. How to compare fluxes in different units, 600 µmol quanta m -2 s -1 v.s. W m -2 nm -1 ?

Thanks, the comment refers to figure 11 that is expressed in µmol quanta m -2 s -1 we propose to correct the text as follows:

Consistent with the results shown in Fig. 11, major discrepancies arose when comparing DPAR, where the model values resulted in much higher values than those obtained from the floats, increasing especially during summer (up to 600 µmol quanta m-2 s-1 in August, as shown in Fig. 13).

6. L331-336. You are effectively saying that for the radiation flux results it is more important that clouds are in correct place in correct time than how the details of liquid cloud optics are treated in the simulated clouds. Which is true, of course.

Thanks for the comment, we propose to add this remark to the text in line 337:

In other terms, this implies that a correct localization of clouds in space and time is more important than specific details of the liquid cloud optics parameterizations.

7. L383-387. See the general comments.

According to the reviewer suggestion we propose to update the conclusions adding the following text:

The atmospheric multispectral input data are necessary to resolve the multispectral propagation of light along the water column. Evaluating the uncertainties and the quality of the these input data is fundamental for all the future applications involving bio-optical modelling and an important starting point to develop assimilation schemes based on bio-optical modelling.

---

## Author Comment (AC3) · 18 Feb 2021

We thank Reviewer #3 for the useful comments. The point by point response is provided below in blue font, and the proposed text modifications in red.

This paper quantifies the skill of the OASIM radiative transfer model in the Mediterranean Sea region. Although there is no (cited) comparison of the skill of other atmospheric optical models, the OASIM performance is impressive, and would be both good news and useful information for the groups using, or considering using, OASIM.
Another interesting feature of the manuscript is the use of the Bio-Argo array for model assessment. As this will no doubt become the primary means of assessing large-scale marine atmospheric models in the future, the technique developed in the manuscript will be interesting to others.
While there are lots of excellent figures, the manuscript itself is quite short and does not make as much of the analysis as it could – leaving some room for additions, which would help to make the manuscript more useful and citable.

A couple of ideas for strengthening the manuscript:
1. The OASIM is assess primarily based on Ed(0-), the downwelling planar irradiance. It is mentioned that the BOUSSOLE observations have above water observations. Analysis of skill of Ed(0+) would help to distinguish between OASIM errors in the atmosphere and in the transmission across the sea-surface interface. This distinction would be useful for both improving OASIM, and also for those using OASIM.

As suggested by the reviewer, we show in Fig. R1 the multispectral downward planar irradiance ($Ed(\lambda,0+)$) simulated by OASIM (blue lines) and measured at BOUSSOLE (red lines) for the month of March. Comparing Fig. R1 with Fig. 6 proposed in the submitted version for $Ed(\lambda,0-)$, we show that differences, related to the atmospheric parameters controlling seas surface reflection, have a low impact on the results. In fact otherwise we would observe a much higher deterioration in model performance when computing $Ed(\lambda,0-)$ from $Ed(\lambda,0+)$. In order to provide a general overview of the differences from $Ed(\lambda,0+)$ from model and from BOUSSOLE we reported the skill in Table R1 in analogy to the Table 1 that we proposed in the manuscript. The differences in the skill (e.g. RMSD and BIAS percentual scores) between computation of $Ed(\lambda,0-)$ from $Ed(\lambda,0+)$ indicates that the model is, in most of the cases, slightly better in computing $Ed(\lambda,0+)$ than $Ed(\lambda,0-)$ but the differences are in any case of second order so is the impact of surface pressure and wind.
We propose to include the following text to the section 3.4 "Summary of the OASIM model skills in the Mediterranean Sea":

> Similar summary analysis performed for $Ed(\lambda,0-)$ was also performed for $Ed(\lambda,0+)$, to estimate the impact on reflection processes at sea atmosphere interface on irradiance (not shown). These processes are regulated by atmospheric parameters shown in Fig. 4. Percentual skill metrics indicate that RMSD is only marginally affected, with differences lower than <1%, while BIAS for $Ed(\lambda,0+)$ shows differences lower than 5% with respect to $Ed(\lambda,0-)$.

[Figure]

Figure R1. Multispectral downward planar irradiance **Ed(λ,0+)** simulated by OASIM (blue lines) and measured at BOUSSOLE (red lines). The wavelengths considered are those measured by the BOUSSOLE sensors for the average March data derived from the time series. For each panel, the reported statistics (RMSD, Bias, r, and regression slope) are related both to the high-frequency signal (with a temporal resolution of 15'; top left) and to the average day in the considered month (top right). The vertical bars indicate the variance in the monthly averaged values of the average day.

Table R1. Summary of the model skill compared to the available data from the BOUSSOLE buoy (from 2004 to 2012) and BGC-Argo floats (from 2012 to 2017) for the irradiance (**Ed(λ,0+)**) at the different wavelengths (WL) and for DPAR. RMSD, bias, and Y-int are expressed in W m-2 nm-1, while all the other indicators (regression R and slope) are dimensionless, where N is the number of match-ups between the model and observations. For the BOUSSOLE comparison, the green numbers are derived by filtering out the day-to-day variability (i.e., the intra-monthly variability). Given the large number of samples, all statistics are significant (p-value < 0.05). For the RMSD and BIAS, the percentage values normalized by average data are reported in parentheses.

| BOUSSOLE vs OASIM-ECMWF [2004-2012] | | | | | | |
|---|---|---|---|---|---|---|
| **WL** | **RMSD** | **BIAS** | **R** | **SLOPE** | **Y-int** | **N** |
| 412.5 | 0.15 (34.1%) | -0.04 (-9.5%) | 0.83 | 0.66 | 0.08 | 55239 |
| | 0.04 (10.0%) | -0.04 (-9.6%) | 0.99 | 0.88 | 0.00 | |
| 442.5 | 0.18 (33.6%) | 0.00 (0.6%) | 0.84 | 0.77 | 0.09 | 111010 |
| | 0.04 (7.2%) | 0.00 (0.5%) | 0.99 | 1.00 | -0.01 | |
| 490 | 0.20 (34.4%) | 0.00 (-0.1%) | 0.84 | 0.76 | 0.10 | 112186 |
| | 0.04 (7.4%) | 0.00 (-0.2%) | 0.99 | 1.00 | -0.02 | |
| 510 | 0.20 (34.6%) | -0.01 (-2.0%) | 0.83 | 0.74 | 0.10 | 112071 |
| | 0.04 (7.2%) | -0.01 (-2.1%) | 0.99 | 0.98 | -0.02 | |
| 555 | 0.20 (33.4%) | 0.03 (5.1%) | 0.85 | 0.83 | 0.10 | 55309 |
| | 0.05 (8.6%) | 0.03 (5.0%) | 0.99 | 1.05 | -0.03 | |
| 560 | 0.20 (35.5%) | 0.01 (2.3%) | 0.83 | 0.76 | 0.11 | 106660 |
| | 0.04 (7.9%) | 0.01 (2.3%) | 0.99 | 1.02 | -0.02 | |
| 665 | 0.18 (34.1%) | -0.02 (-3.0%) | 0.84 | 0.75 | 0.09 | 76247 |
| | 0.04 (7.1%) | -0.02 (-3.1%) | 0.99 | 0.99 | -0.03 | |
| 670 | 0.17 (39.6%) | -0.04 (-9.3%) | 0.79 | 0.63 | 0.08 | 32733 |
| | 0.04 (10.1%) | -0.04 (-9.5%) | 0.98 | 0.92 | -0.02 | |
| 681.25 | 0.17 (36.4%) | -0.08 (-16.4%) | 0.81 | 0.62 | 0.07 | 110418 |
| | 0.05 (10.3%) | -0.07 (-16.6%) | 0.99 | 0.85 | -0.02 | |

2. Following on from 1., the article distinguishes between errors in cloud-free and cloudy days. I wonder whether a somewhat similar, but more diagnostically-useful distinction might be the fractions of direct and diffuse radiation. As per point 1 this might be more useful for OASIM developers and also for those attempting to apply these results outside the Mediterranean where lower sun angles might change the balance of direct vs diffuse for the same cloud.

Thanks for the comment. We introduced the following indicator:

$$\mathbf{IND=Edir(\lambda,0\text{-})/(Edir(\lambda,0\text{-})+Edif(\lambda,0\text{-}))*100} \quad (R1)$$

**IND** varies in the interval [0,100]. **IND** is 0 when $\mathbf{Ed(\lambda,0\text{-}) = Edif(\lambda,0\text{-})}$ and it is 100 when $\mathbf{Ed(\lambda,0\text{-}) = Edir(\lambda,0\text{-})}$. **IND** =50 indicates perfect balance: $\mathbf{Edir(\lambda,0\text{-}) = Edif(\lambda,0\text{-})}$.

[Figure]

Figure R2. Comparison of the OASIM downward irradiance ($Ed(\lambda,0\text{-})$ at the nine wavelengths) to the BOUSSOLE data from 2004 to 2012 in terms of the RMSD and regression slope and their relationship with the ECMWF ERA-Interim cloud cover (CC). The left section of each panel shows the monthly climatology of the RMSD (normalized by its averaged value; red lines and labels) and regression slope (normalized by its averaged value; black line), superimposed on the monthly climatology of the cloud cover (in %, blue bars and labels). Regression slope thresholds at 1 (dotted black line) and 0.75 (dot-dashed black line) are also shown. The right section of each panel shows the monthly means of the time series of the RMSD (red dots, with a 0.5 value; red dotted line), superimposed on the monthly means of the time series of the cloud cover (blue bars and labels). The IND parameter defined in equation 3 is also reported (cyan lines), and in all panels varies in the range [0,100]

We complemented Figure R2 (corresponding to Figure 14 in the manuscript) including the indicator **IND** in cyan defined above. The indicator shows higher values during summer, when the direct component is dominant, corresponding to better model performances. On the contrary, during winter lower values of **IND** are found, with the diffuse component

dominating, corresponding to the worser model performances. Similar results are observed considering the data corresponding to the Mediterranean Sea region as reported in Fig. R3.

[Figure]

Figure R3. Evaluation of the IND indicator normalized to 1, for the three wavelengths measured by BGC Argo floats panel a) **λ**=380 nm, panel b) **λ** = 412 nm, panel c) **λ** = 490 nm. Panel d) shows the cloud cover.

We propose to add the following text to the discussion section:
 Beside cloud cover, we introduced a further diagnostic (IND) based on the fraction of direct and diffuse irradiance components.
This diagnostic is defined as:

$$\textbf{IND} = \textbf{Edir}(\boldsymbol{\lambda},\textbf{0-})/(\textbf{Edir}(\boldsymbol{\lambda},\textbf{0-})+\textbf{Edif}(\boldsymbol{\lambda},\textbf{0-}))*100 \quad (3)$$

where IND is a-dimensional and varies in the interval [0,100]. Recalling that **Ed(λ,0-)** = **Edir(λ,0-)** + **Edif(λ,0-)**, **I** is 0 when **Ed(λ,0-)** = **Edif(λ,0-)** and it is 100 when **Ed(λ,0-)** = **Edir(λ,0-)**. **I** =50 indicates perfect balance: **Edir(λ,0-)** = **Edif(λ,0-)**. As show in Fig. 14, IND provides similar interpretation of cloud cover in fact the model skill is higher when IND is higher and vice-versa. This diagnostic indicator could be used to generalize results in regions outside the Mediterranean Sea where lower sun angles are found implying a different balance of the direct versus the diffuse component. In these situations the effect of clouds in increasing RMSD and bias could be even higher. It is worth to mention that in the present case, since IND covariates with cloud cover, it is difficult to separate the role of clouds from direct versus diffuse irradiance ratio. Nonetheless, IND at 412 nm is lower than all the other wavelengths and this could explain, at least in part, the lower skill observed at 412 nm.

3. The extrapolation of the Bio-Argo data to the surface sounds like a critical step. Can you give more details? For example, do you assume an exponential decay with depth?
Thanks for pointing this out, we propose to rephrase with the following text:
Before comparing model values to observations, the irradiance profiles obtained from floats were extrapolated to the surface with an exponential fitting procedure based on the curve_fit tool of the python package scipy. Further, we required profiles to have at least one measurement in the first 1.5 m depth from sea surface and to have at least 4 measurements in

the first 10 m. In addition, any sub-basin (as defined in Fig.2) and month containing fewer than 5 profiles was discarded

Minor comments.

1. L23 "daily variability" – this is somewhat ambiguous. Do you mean between days or within days?
   We mean between days variability we substitute "when the daily variability is filtered out." with the following: "when the variability between days is filtered out."

2. L24 replace 'high' with a number. We propose to rephrase with "Both BOUSSOLE and BGC-Argo indicate that bias is up to 20 % for the irradiance at 380 nm, 412 nm, and for wavelenghts higher than 670 nm, whereas it decreases to less than 5% at the other wavelengths."

3. 'cloud dynamics and seasonality" – the former is a process, the later a timescale. They don't quite fit together within one phrase. We agree, we removed "seasonality".

4. L30 remove "notably" Ok, removed.

5. L47 -51. Confusing. Tried to say too much in one sentence. We propose to rephrase in order to make the sentence more clear. This meets the requirements and high data quality standard expected for remote system calibration of ocean colour spaceborne sensors (Antoine et al., 2020) and for the Copernicus biogeochemical operational model system for the Mediterranean Sea (MedBFM; Lazzari et al., 2010, 2012, 2016; Cossarini et al., 2015; Teruzzi et al., 2014, 2018, 2019; Salon et al., 2019). This system is used to produce analysis, forecasts and reanalysis of the biogeochemical state, recently upgraded to assimilate BGC-Argo float data (Cossarini et al., 2019).

6. L83 'while' – this is not a 'while' we propose to change the sentence as follows: "In OASIM, gaseous absorption by ozone, oxygen, carbon dioxide and water vapor is resolved before cloud transmittance determination, and aerosol effects are ignored in the presence of clouds."

7. L84 – remove 'the' before 'aerosol'. Ok, we agree.

8. L107 "resolve the diurnal variability" – this is within a day right? Yes, it is.

9. L108 'properly' is a subjective, rather than objective, adjective. We remove this adjective.

10. L115 remove 'totally'. Ok, removed.

11. L134. At this point I didn't know how you were defining regions. We rephrased the text as indicated above specifying that the sub-basins are shown in Fig.2.

12. L166. Would it be more accurate to say "W m-2 per waveband"? Yes, we propose the updated the text with "The OASIM outputs for the irradiance are expressed in W m-2 per waveband"

13. L167 "to W m-2". We converted the data and model to W m-2 nm-1 as shown in the figures.

14. L202 – 15' and 1-degree – most people would write 15 minutes and 1o!. Yes, we updated to suggested standards.

15. L257 – stick with bias = model – observation. Don't say a negative bias, but a bias of -20%. Otherwise it can get confusing. Yes, we agree, we changed the text accordingly.

16. L264, L265 – subscript of d different to elsewhere. Ok, we corrected.

17. L294 'float cluster" is new wording that is unnecessary. We modified with BGC-Argo float.

18. L354. More details on the wavelength discretisation. For example, Does 412 sit in a 400 to 425 nm band? The bands are centred on 400nm and 425nm so 412 nm is at the interface of two bands. We propose to add the following sentence to the text at L 354: "Especially due to the fact that in the present simulations the 412 nm wavelength is at the interface between the band centred at 400 nm and the one centred at 425 nm.".

19. L373 "The OASIM model . . . this information" – Conclusions need tighter sentence than this. We agree to remove this sentence.

20. L386-387 – this manuscript could be more helpful for motivating this sentiment as per main point 2. We propose to add the following sentence to the conclusions: L385:" The improvement of the model skill at BOUSSOLE, when variability between days is filtered out, indicates that spatial and temporal resolution in resolving clouds distributions is probably the most important parameter affecting skill. Nonetheless, the analysis of the relative contribution of Edir and Edif indicates that skill is correlated to their ratio, suggesting that improving the physical description of the radiative processes should be considered. To this end, novel atmospheric models …"

21. Fig 4. Wind speed at what height? Wind speed is at 10 meters, we updated the caption of Fig. 4 including also this information.

22. Fig 12-Fig 13 – It took me a moment to work out what 'B' and 'F' meant, especially since you them both BioArgos and floats. In figure BOUSSOLE and BioArgo would be quicker for interpretation. Yes, we agree with the reviewer to write the name in full extensions.

---

## Author Response (AR2)

Dear Editor
all the suggestions have been considered and the manuscript updated.

Best Regards

Paolo Lazzari on behalf of the co-Authors.

Comments to the Author:
I thank the authors for taking up the suggestions of the reviewers. The paper is now considerably easier to understand as a result. I do, however, have a few additional comments, most of which relate to language. I do not need to see another version of the manuscript.

1. Lines 43-44: I suggest rewriting as "…at the ocean-atmosphere interface is important both for the solution of the radiative transfer model within the water column and for the development of…" DONE

2. Line 142: "from the topmost 10-m ocean layer", not "of" DONE

3. Lines 163-165: These are essentially a repeat of lines 149-150. These could perhaps be deleted, and the next sentence rewritten to begin "Data analysis covered the time period shown in Fig.3, using datasets from…" DONE

4. Line 233: "minute" not "minutes" in both places. DONE, same at Line 209, 217

5. Line 346: "In other words…" not "other terms" DONE

6. Lines 362-363: "It is worth mentioning that in the present case, since IND covaries with…" DONE

7. Line 400: "within the water column" DONE

8. In response to reviewer 2, you stated that you would add the references to the work of Bozzo et al. and Rontu et al., but I don't see these anywhere in the text nor in the reference list. DONE

9. In response to reviewer 3, point 20, you stated that you would include the sentence "The improvement of the model skill at BOUSSOLE…is probably the most important parameter affecting skill." Yet you don't say this explicitly. As this is a really important point, perhaps you could rewrite lines 466-467 to accentuate this, for example: "The coarse resolution of cloud cover likely affects the model skill at both seasonal and intra-monthly time-scales. This is shown by the improvement in model skill at BOUSSOLE when variability between days is filtered out." DONE